# CONTRASTIVE PREFERENCE LEARNING: LEARNING FROM HUMAN FEEDBACK WITHOUT RL

**Joey Hejna**
Stanford University
jhejna@cs.stanford.edu

**Rafael Rafailov** *
Stanford University
rafailov@cs.stanford.edu

**Harshit Sikchi** *
UT Austin
hsikchi@utexas.edu

**Chelsea Finn**
Stanford University

**Scott Niekum**
UMass Amherst

**W. Bradley Knox**
UT Austin

**Dorsa Sadigh**
Stanford University

## ABSTRACT

Reinforcement Learning from Human Feedback (RLHF) has emerged as a popular paradigm for aligning models with human intent. Typically RLHF algorithms operate in two phases: first, use human preferences to learn a reward function and second, align the model by optimizing the learned reward via reinforcement learning (RL). This paradigm assumes that human preferences are distributed according to reward, but recent work suggests that they instead follow the *regret* under the user's optimal policy. Thus, learning a reward function from feedback is not only based on a flawed assumption of human preference, but also leads to unwieldy optimization challenges that stem from policy gradients or bootstrapping in the RL phase. Because of these optimization challenges, contemporary RLHF methods restrict themselves to contextual bandit settings (e.g., as in large language models) or limit observation dimensionality (e.g., state-based robotics). We overcome these limitations by introducing a new family of algorithms for optimizing behavior from human feedback using the *regret*-based model of human preferences. Using the principle of maximum entropy, we derive Contrastive Preference Learning (CPL), an algorithm for learning optimal policies from preferences without learning reward functions, circumventing the need for RL. CPL is fully off-policy, uses only a simple contrastive objective, and can be applied to arbitrary MDPs. This enables CPL to elegantly scale to high-dimensional and sequential RLHF problems while being simpler than prior methods.

## 1 INTRODUCTION

As large pretrained models have become increasingly performant, the problem of aligning them with human preferences has risen to the forefront of research. This alignment is especially difficult when larger datasets inevitably include suboptimal behaviors. Reinforcement learning from human feedback (RLHF) has emerged as a popular solution to this problem. Using human preferences, RLHF techniques discriminate between desirable and undesirable behaviors with the goal of refining a learned policy. This paradigm has shown promising results when applied to finetuning large language models (LLMs) (Ouyang et al., 2022), improving image generation models (Lee et al., 2023), and adapting robot policies (Christiano et al., 2017) – all from suboptimal data. For most RLHF algorithms, this process includes two phases. First, a reward model is trained from user preference data. And second, that reward model is optimized by an off-the-shelf reinforcement learning (RL) algorithm.

Unfortunately, this two-phase paradigm is founded on a flawed assumption. Algorithms that learn reward models from preference data require that human preferences are distributed according to the discounted sum of rewards or *partial return* of each behavior segment. However, recent work (Knox et al., 2022) calls this into question, positing that humans instead provide preferences based on the *regret* of each behavior under the optimal policy of the expert's reward function. Intuitively, a human's judgement is likely based on optimality, instead of which states and actions have higher quantity for

---

*Equal Contribution.     Our code is released at https://github.com/jhejna/cpl

reward. As a result, the correct quantity to learn from feedback might not be the reward, but instead the optimal *advantage* function or, in other words, the negated regret.

In their second phase, two-phase RLHF algorithms optimize the reward function learned from the first phase with RL. In practice, RL algorithms suffer from a suite of optimization challenges stemming from temporal credit assignment, such as the high-variance of policy gradients (Marbach & Tsitsiklis, 2003) or instability of approximate dynamic programming (Van Hasselt et al., 2018). Thus, past works limit their scope to circumvent these issues. For instance, RLHF techniques for LLMs assume a contextual bandit formulation (Ouyang et al., 2022), where the policy receives a single reward value in response to a given query to the user. While this reduces the need for long-horizon credit assignment, and consequently the high variance of policy gradients, in reality user interactions with LLMs are multi-step and sequential, violating the single-step bandit assumption. As another example, RLHF has been applied to low-dimensional state-based robotics problems (Christiano et al., 2017; Sikchi et al., 2023a), a setting where approximate dynamic programming excels, but not yet scaled to more realistic high-dimensional continuous control domains with image inputs. Broadly, RLHF methods not only incorrectly assume that the reward function alone drives human preferences, but also require mitigating the optimization challenges of RL by making restrictive assumptions about the sequential nature of problems or dimensionality.

In this work, we introduce a new family of RLHF methods that use a *regret*-based model of preferences, instead of the commonly accepted partial return model that only considers the sum of rewards. Unlike the partial return model, the regret-based model directly provides information about the optimal policy. A fortunate outcome of this is that it completely eliminates the need for RL, allowing us to solve RLHF problems in the general MDP framework with high-dimensional state and action spaces. Our key insight is to combine the *regret*-based preference framework with the principle of Maximum Entropy (MaxEnt), resulting in a bijection between advantage functions and policies. By exchanging optimization over advantages for optimization over policies, we are able to derive a purely supervised learning objective whose optimum is the optimal policy under the expert's reward. We refer to our approach as Contrastive Preference Learning due to its resemblance with commonly accepted contrastive learning objectives.

CPL has three key benefits over prior work. First, CPL can scale as well as supervised learning because it uses *only supervised objectives* to match the optimal advantage without any policy gradients or dynamic programming. Second, CPL is *fully off-policy*, enabling effectively using any offline sub-optimal data source. Finally, CPL can be applied to *arbitrary Markov Decision Processes* (MDPs), allowing for learning from preference queries over sequential data. To our knowledge, no prior methods for RLHF simultaneously fulfill all three of these tenets. To demonstrate CPL's adherence to the three aforementioned tenets, we show its effectiveness on sequential decision making problems with sub-optimal and high-dimensional off-policy data. Notably, we show that CPL can effectively use the same RLHF fine tuning procedure as dialog models to learn temporally extended manipulation policies in the MetaWorld Benchmark. Specifically, we pretrain policies using supervised learning from high-dimensional image observations, before fine tuning them with preferences. Without dynamic programming or policy gradients, CPL is able to match the performance of prior RL based methods. At the same time, it is $1.6\times$ faster and four times as parameter efficient. When using denser preference data, CPL is able to surpass the performance of RL baselines on 5 out of 6 tasks.

## 2 PRELIMINARIES

We consider the general reinforcement learning from human feedback (RLHF) problem within a reward-free MDP $\mathcal{M}/r = (\mathcal{S}, \mathcal{A}, p, \gamma)$ with state space $\mathcal{S}$, action space $\mathcal{A}$, transition dynamics $p(s_{t+1}|s_t, a_t)$, and discount factor $\gamma$. We assume all states are reachable by some policy. The goal of RLHF is to learn a policy $\pi(a|s)$ that maximizes an expert user's reward function $r_E(s, a)$. However, since the reward function is not given in an MDP $/r$, it must be inferred from the expert's preferences. Typically, a user preference orders two behavior segments. A length-$k$ segment is denoted $\sigma = (s_1, a_1, s_2, a_2, ..., s_k, a_k)$. We use $\sigma^+ \succ \sigma^-$ to indicate that segment $\sigma^+$ was preferred to $\sigma^-$ by the user without loss of generality and assume we are given a dataset $\mathcal{D}_{\text{pref}} = \{(\sigma_i^+, \sigma_i^-)\}_{i=1}^n$ of such preferences where $\sigma^+ \succ \sigma^-$.

**Maximum Entropy Reinforcement Learning**. The aim of *maximum-entropy reinforcement learning* is to learn a policy $\pi$ that maximizes its causal entropy in addition to the cumulative discounted return:

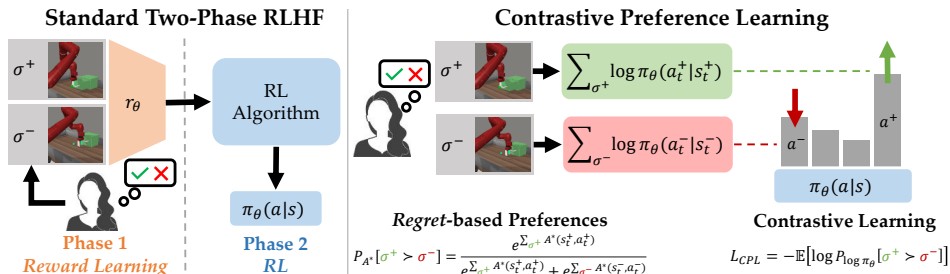

Figure 1: While most RLHF algorithms use a two-phase reward learning, then RL approach, CPL directly learns a policy using a contrastive objective. This is enabled by the regret preference model.

$$\max_{\pi} \mathbb{E}_{\pi}\left[\sum_{t=0}^{\infty}\gamma^t(r(s_t,a_t)-\alpha\log\pi(a_t|s_t))\right], \tag{1}$$

where $\alpha$ is a temperature parameter. Augmenting the reward function with an additional negated $\log\mu(a|s)$ term for reference distribution $\mu(a|s)$ yields the KL-constrained objective used in offline RL (Levine & Koltun, 2013; Garg et al., 2023) and prominent RLHF approaches for LLMs (Ziegler et al., 2019; Ouyang et al., 2022). Though we adopt the standard maximum entropy framework, our approach easily extends to the constrained setting. Under policy $\pi$ and reward function $r$, we denote the state-value function by $V_r^{\pi}(s)$ and state-action value function by $Q_r^{\pi}(s,a)$. The advantage function, $A_r^{\pi}(s,a) \triangleq Q_r^{\pi}(s,a) - V_r^{\pi}(s)$, measures how much *worse* taking action $a$ is than acting according to $\pi$. We use $\pi^*$ as short-hand for the solution to Eq. (1) with reward function $r_E$, and write its corresponding corresponding value functions as $V^*(s)$ and $Q^*(s,a)$ instead of $V_{r_E}^{\pi^*}$ and $Q_{r_E}^{\pi^*}$. We measure the optimality of behavior directly by using the advantage function of $\pi^*$, $A^*(s,a)$.

**The Regret (or Advantage) Preference Model.** Learning $\pi^*$ requires characterizing how preferences are generated according to a preference model $P_E[\sigma^+ \succ \sigma^-]$, or the probability the expert prefers $\sigma^+$ to $\sigma^-$. Typically, the preference model is chosen to be the Boltzmann rational distribution over each segment's discounted partial return, $\sum_{t=1}^{k}\gamma^t r_E(s_t,a_t)$, where $r_E$ is the expert's hidden reward function. However, such models have been shown to be inconsistent with real human preferences (Knox et al., 2022). For instance, consider a sparse reward $r_E(s,a) = 1\{s=g\}$. Two segments that do not reach the goal would have the same partial returns even if one moved towards the goal $g$ while the other moved away from it. This inconsistency is resolved by considering preferences to be distributed according to the Boltzmann rational distribution over the negated discounted *regret* under $r_E$, or $-\sum_{t=1}^{k}\gamma^t(V^*(s_t) - Q^*(s_t,a_t))$. In this framework, a user's preference indicates that a segment has lower regret with respect to their intended optimal policy. Leveraging the equivalence of negated regret and the discounted sum of optimal advantages, we equivalently write the regret-based preference model as

$$P_{A^*}[\sigma^+ \succ \sigma^-] = \frac{\exp\sum_{\sigma^+}\gamma^t A^*(s_t^+,a_t^+)}{\exp\sum_{\sigma^+}\gamma^t A^*(s_t^+,a_t^+) + \exp\sum_{\sigma^-}\gamma^t A^*(s_t^-,a_t^-)}, \tag{2}$$

where we use the shorthand "$+$" and "$-$" as indexing the states and actions of segments $\sigma^+$ and $\sigma^-$. In the next section, we use the regret preference model in combination with the principle of maximum causal entropy to derive CPL.

## 3 CONTRASTIVE PREFERENCE LEARNING

Though recent work has shown that human preferences are better modeled by the optimal advantage function or regret, most existing RLHF algorithms assume otherwise. By learning a reward function with a mistaken model of preference and then applying RL, traditional RLHF approaches incur a vast, unnecessary computational expense (Knox et al., 2023). Our aim is to derive simple and scalable RLHF algorithms that are purpose-built for the more accurate regret model of human preferences.

Modeling human preferences with regret is not new, but past work suffers from a number of shortcomings. Specifically, existing algorithms using the regret preference model are brittle, as they rely on estimating gradients with respect to a moving reward function, which thus far has only been approximated by computing successor features and assuming a correct linear or tabular representation of the expert reward function $r_E$ (Knox et al., 2022; 2023). Consequently, these algorithms appear unsuitable for complex scenarios beyond the simplistic grid world environments in which they have been tested.

The key idea of our approach is simple: we recognize that the advantage function, used in regret preference model, can easily be replaced with the log-probability of the policy when using the maximum entropy reinforcement learning framework. The benefit of this simple substitution is however immense. Using the log-probability of the policy circumvents the need to learn the advantage function or grapple with optimization challenges associated with RL-like algorithms. In sum, this enables us to not only embrace a more closely aligned regret preference model, but also to exclusively rely on *supervised learning* when learning from human feedback.

In this section, we first derive the CPL objective and show that it converges to the optimal policy for $r_E$ with unbounded data. Then, we draw connections between CPL and other supervised-learning approaches. Finally, we provide recipes for using CPL in practice. Our algorithms are the first examples of a new class of methods for sequential decision making problems which directly learn a policy from regret based preferences without RL, making them far more efficient.

## 3.1 FROM OPTIMAL ADVANTAGE TO OPTIMAL POLICY

Under the regret preference model, our preference dataset $\mathcal{D}_{\text{pref}}$ contains information about the optimal advantage function $A^*(s,a)$, which can intuitively be seen as a measure of how much *worse* a given action $a$ is than an action generated by the optimal policy at state $s$. Therefore, actions that maximize the optimal advantage are by definition an optimal actions and learning the optimal advantage function from preferences should intuitively allow us to extract the optimal policy.

**Naïve approach.** When presented with $\mathcal{D}_{\text{pref}}$, one might naïvely follow the standard RLHF reward modeling recipe, but with advantages. This would equate to optimizing a parameterized advantage $A_\theta$ to maximize the log likelihood of $\mathcal{D}_{\text{pref}}$ given the preference model in Eq. (2), or $\max_{A_\theta} \mathbb{E}_{(\sigma^+,\sigma^-)\sim\mathcal{D}_{\text{pref}}}[\log P_{A_\theta}[\sigma^+ \succ \sigma^-]]$, where $P_{A_\theta}$ is the preference model induced by the learned advantage function. Once an advantage function that aligns with the preference data is learned, it could be distilled into a parameterized policy. At first glance, it seems like this simple two-step approach could be used to recover the optimal policy from preference data. However, it turns out that learning a Bellman-consistent advantage function is non-trivial in both standard and MaxEnt RL, making learning a valid intermediate advantage function not only unnecessary, but also *harder* in practice.

**Eliminating the need to learn advantage.** In maximum entropy RL, Ziebart (2010) has shown that the following relationship between the optimal advantage function and optimal policy holds:

$$\pi^*(a|s) = e^{A_r^*(s,a)/\alpha}.$$

This means that in order for a learned advantage function to be optimal, it must be normalized, that is $\int_{\mathcal{A}} e^{A^*(s,a)/\alpha} da = 1$. Enforcing this constraint is intractable, particularly in continuous spaces with large neural networks, making naïvely learning $A_\theta$ via maximum likelihood estimation difficult.

However, one might instead notice that the above equation establishes a bijection between the advantage function $A_r^*$ and the policy $\pi^*$, namely that the optimal advantage function is proportional to the optimal policy's log-likelihood:

$$A_r^*(s,a) = \alpha\log\pi^*(a|s). \tag{3}$$

This means that instead of learning the optimal advantage function, we can directly learn the optimal policy. Given preferences are distributed according to the optimal advantage function for the expert reward function $r_E$, we can write the preference model in terms of the optimal policy $\pi^*$ by substituting Eq. (3) into Eq. (2) as follows,

$$P_{A^*}[\sigma^+ \succ \sigma^-] = \frac{\exp\sum_{\sigma^+}\gamma^t\alpha\log\pi^*(a_t^+|s_t^+)}{\exp\sum_{\sigma^+}\gamma^t\alpha\log\pi^*(a_t^+|s_t^+) + \exp\sum_{\sigma^-}\gamma^t\alpha\log\pi^*(a_t^-|s_t^-)}. \tag{4}$$

Thus, the maximum entropy framework has led to a model of human preferences that is solely in terms of the optimal policy $\pi^*$. Using this equivalent form of the advantage-based preference model, we can directly optimize a learned policy $\pi_\theta$ to match the preference model via maximum likelihood with the following convex objective:

$$\mathcal{L}_{\text{CPL}}(\pi_\theta,\mathcal{D}_{\text{pref}}) = \mathbb{E}_{(\sigma^+,\sigma^-)\sim\mathcal{D}_{\text{pref}}}\left[-\log\frac{\exp\sum_{\sigma^+}\gamma^t\alpha\log\pi_\theta(a_t^+|s_t^+)}{\exp\sum_{\sigma^+}\gamma^t\alpha\log\pi_\theta(a_t^+|s_t^+) + \exp\sum_{\sigma^-}\gamma^t\alpha\log\pi_\theta(a_t^-|s_t^-)}\right]. \tag{5}$$

Assuming sufficient representation power, at convergence $\pi_\theta$ will perfectly model the user's preferences, and thus exactly recover $\pi^*$ under the advantage-based preference model given an unbounded amount of preference data. Specifically, in Appendix B, we prove the following Theorem:

**Theorem 1.** *Assume an unbounded number of preferences generated from a noisy rational regret-preference model with expert advantage function $A^*$. CPL recovers the optimal policy $\pi^*$ corresponding to reward $r_E$.*

This proof relies on the bijection between optimal advantage functions and policies in maximum entropy RL and the fact that the regret preference model is *identifiable* (Knox et al., 2022), meaning the objective can achieve a loss of zero.

**Benefits of directly learning the policy.** Directly learning $\pi$ in this manner has several benefits, both practical and theoretical. Perhaps most obviously, directly learning the policy circumvents the need for learning any other functions, like a reward function or value function. This makes CPL extremely simple in comparison to prior work. When scaling to larger models, only learning the policy reduces both complexity and computational cost. Second, as pointed out by prior works (Christiano et al., 2017; Hejna & Sadigh, 2023), reward learning can be harmed by the invariance of Boltzmann rational preference models (Eq. (2)) to shifts; i.e., adding a constant to each exponent does not change $P[\sigma^+ \succ \sigma^-]$. In CPL the distributional constraint of the policy ($\pi_\theta(a|s) \geq 0$ for all $a$ and $\int_{\mathcal{A}} \pi_\theta(a|s)da = 1$) remedies this issue automatically, since adding a constant makes $\int_{\mathcal{A}} \pi_\theta(a|s)da \neq 1$. Finally, per previous arguments, the policy's distributional constraint guarantees that $\int_{\mathcal{A}} e^{A_\theta(s,a)/\alpha}da = 1$. Thus, it can be shown that CPL's learned implicit advantage function is *always* the optimal advantage function for some reward function. We call this property, defined below, *consistency* and prove the following Proposition in Appendix B.

**Definition 1.** *An advantage function $A(s,a)$ is consistent if there exists some reward function $r(s,a)$ for which $A$ is the optimal advantage, or $A(s,a) = A_r^*(s,a)$.*

**Proposition 1.** *CPL learns a consistent advantage function.*

The consequences of this are that no matter the amount of preference data used, CPL will always learn the optimal policy for *some* reward function, and adding additional preference data only improves the implicit estimate of $r_E$.

**Connections to Contrastive Learning.** When deriving CPL, we intentionally chose to denote preferred and unpreferred behavior segments by "+" and "-" to highlight the similarities between CPL and contrastive learning approaches. Though some two-phase RLHF approaches have drawn connections between their reward learning phase and contrastive learning (Kang et al., 2023), CPL directly uses a contrastive objective for policy learning. Specifically, Eq. (5) is an instantiation of the Noise Constrastive Estimation objective (Gutmann & Hyvärinen, 2010) where a segment's score is its discounted sum of log-probabilities under the policy, the positive example being $\sigma^+$ and the negative $\sigma^-$. In the appendix we show that when applied to ranking data using a Plackett-Luce Model, CPL recovers the InfoNCE objective from Oord et al. (2018) where the negative examples are all the segments ranked below the positive segment. Effectively, CPL has fully exchanged the reinforcement learning objective for a supervised, representation learning objective while still converging to the optimal policy. As marked success has been achieved applying contrastive learning objectives to large-scale datasets and neural networks (Chen et al., 2020; He et al., 2020; Radford et al., 2021), we expect CPL to scale more performantly than RLHF methods that use traditional RL algorithms.

## 3.2 PRACTICAL CONSIDERATIONS

The Contrastive Preference Learning framework provides a general loss function for learning policies from advantage-based preferences, from which many algorithms can be derived. In this section, we detail practical considerations for one particular instantiation of the CPL framework which we found to work well in practice. In the appendix, we include several instantiations of CPL for different types of data and conservative regularizers.

**CPL with Finite Offline Data.** Though CPL converges to the optimal policy with unbounded preference data, in practice we are often interested in learning from finite offline datasets. In this setting, policies that extrapolate too much beyond the support of the dataset perform poorly as they take actions leading to out of distribution states. Like many other preference-based objectives, CPL's objective is not *strictly* convex (Appendix B.3). Thus, many policies, even those with a high weight on actions not in the dataset, can achieve the same optima of Eq. (5). We demonstrate this by formulating CPL as a logistic regression problem. Let the policy be represented by a one-dimensional vector $\pi \in \mathbb{R}^{|\mathcal{S} \times \mathcal{A}|}$. The difference between positive and negative segments, $\sum_{\sigma^+} \gamma^t \alpha \log \pi_\theta(a_t^+|s_t^+) - \sum_{\sigma^+} \gamma^t \alpha \log \pi_\theta(a_t^-|s_t^-)$ can be re-written as a dot-product between $\pi$ and a "comparison" vector $x$, whose values are either

$\gamma^t$, $-\gamma^t$, or $0$ indicating membership to the comparison $\sigma^+ \succ \sigma^-$. Using the logistic function, $\text{logistic}(z) = \frac{1}{1+e^{-z}}$, we re-write the CPL objective in the finite case as

$$\mathcal{L}_{\text{CPL}}(\pi_\theta, \mathcal{D}_{\text{pref}}) = -\sum_{i=1}^{|\mathcal{D}_{\text{pref}}|} \log \text{logistic}(\alpha x_i^\top \log \pi(a|s)), \text{ where } x_i[s,a] = \begin{cases} \gamma^t & \text{if } \sigma_{i,t}^+ = (s,a) \\ -\gamma^t & \text{if } \sigma_{i,t}^- = (s,a) \\ 0 & \text{otherwise} \end{cases}$$

where $\sigma_{i,t}^+$ denotes the $t$th timestep of the preferred segment from the $i$th comparison in $\mathcal{D}_{\text{pref}}$. We can reason about the set of all policies that yield the same CPL loss by assembling all comparison vectors into a matrix $X$, where the $i$th row of $X$ is the vector $x_i$ for the $i$th comparison in the dataset. Any changes to $\log \pi$ in the null space of $X$ have no effect on the logits of the logistic function, and consequently no effect on the loss. In practice, $|\mathcal{S} \times \mathcal{A}| >> n$, making the null space of $X$ often nontrivial such that there are multiple minimizers of the CPL loss, some of which potentially place a high probability on state-action pairs not in the dataset. In Appendix B.3 we provide constructions of $X$ where this is true. Next, we show how this problem can be resolved by incorporating regularization into the CPL objective.

**Regularization.** In finite settings, we want to choose the policy that minimizes the CPL loss function while placing higher likelihood on actions in the dataset. To accomplish this, we modify Eq. (5) with a conservative regularizer that assigns *lower* loss when the policy has *higher* likelihood on actions in $\mathcal{D}_{\text{pref}}$, keeping it in-distribution. Though there are many possible choices of regularizers, we use an asymmetric "bias" regularizer adapted from An et al. (2023) as it performed best in our experiments. Within our objective, the bias regularizer down-weights negative segments by $\lambda \in (0,1)$ as so:

$$\mathcal{L}_{\text{CPL}(\lambda)}(\pi_\theta, \mathcal{D}_{\text{pref}}) = \mathbb{E}_{\mathcal{D}_{\text{pref}}} \left[ -\log \frac{\exp \sum_{\sigma^+} \gamma^t \alpha \log \pi_\theta(a_t^+|s_t^+)}{\exp \sum_{\sigma^+} \gamma^t \alpha \log \pi_\theta(a_t^+|s_t^+) + \exp \lambda \sum_{\sigma^-} \gamma^t \alpha \log \pi_\theta(a_t^-|s_t^-)} \right]. \quad (6)$$

If the policy places more weight on actions in the dataset, $\log \pi_\theta(a|s)$ will increase. In the standard Boltzmann model, increasing the log-probabilities of both the positive and negative segments by the same amount would have no effect on the loss. The bias, however, weighs the increased log-probabilities of the negative segments less, which ultimately decreases the loss. Thus, while a minimizer of the vanilla CPL loss function could place a high probability on unseen actions, Eq. (6) is minimized with a higher weight on in-distribution actions. This is formally captured by the following proposition, which shows that, for a fixed policy, $\mathcal{L}_{\text{CPL}(\lambda)}$ is lower when the policy places a higher likelihood on actions in the dataset versus other comparisons with the same CPL Loss.

**Proposition 2.** *Consider a comparison $\sigma^+ \succ \sigma^-$ from $\mathcal{D}_{pref}$ and an arbitrary comparison $\sigma'^+ \succ \sigma'^-$ such that $\mathcal{L}_{CPL}(\pi, \sigma^+ \succ \sigma^-) = \mathcal{L}_{CPL}(\pi, \sigma'^+ \succ \sigma'^-)$ for a fixed policy $\pi$. If $\sum_{\sigma^+} \gamma^t \log \pi(a_t^+|s_t^+) > \sum_{\sigma'^+} \gamma^t \log \pi(a_t^+|s_t^+)$, then $\mathcal{L}_{CPL(\lambda)}(\pi, \sigma^+ \succ \sigma^-) < \mathcal{L}_{CPL(\lambda)}(\pi, \sigma'^+ \succ \sigma'^-)$.*

Essentially, this shows that the bias regularizer breaks ties in the CPL loss function by penalizing lower likelihoods. We prove this, along with a more general version, in Appendix B.4. In Appendix C we also consider CPL variants with other forms of conservative regularization.

**Pretraining.** Pre-training the policy $\pi_\theta$ with behavior cloning (BC) is a common practice in RLHF (Ouyang et al., 2022). Thus, before fine-tuning with preferences using the CPL loss, we trained the policy using the standard maximum likelihood BC objective, $\min_\theta \mathbb{E}_{(s,a) \sim \mathcal{D}}[\log \pi_\theta(a|s)]$. Though $\mathcal{D}$ could be any dataset, we chose $\mathcal{D}_{\text{pref}}$. We found that this helped performance in some cases, but hurt it others (Appendix D). We posit that pre-training helps when a policy closer to the data distribution is desirable, particularly when out-of-distribution actions are detrimental.

## 4 EXPERIMENTS

In this section, we address the following questions about CPL: First, is CPL effective at fine-tuning policies from regret-based preferences? Second, does CPL scale to high-dimensional control problems and larger networks? Third, what ingredients of CPL are important for attaining high performance? Additional experiments with human data and details are included in the appendix.

**Preference Data.** We evaluate CPL's ability to learn policies for general MDPs from *sub-optimal off-policy rollout data* and preferences. In particular, we consider the training procedure commonly used for large foundation models: supervised learning followed by fine-tuning with RLHF. To do this, we use six tasks from the MetaWorld robotics benchmark (Yu et al., 2020). First, we train baseline policies until they approximately reach a 50% success rate. Then, we rollout 2500 episodes for each suboptimal

stochastic policy. We then form synthetic preference datasets $\mathcal{D}_{\text{pref}}$ of different sizes by sampling segments of length 64 uniformly from the rollout data. We estimate regret-based preference labels using the $Q$-function and policy of an oracle Soft Actor-Critic (SAC) (Haarnoja et al., 2018) model trained to 100% success on a combination of the suboptimal rollout and online data. In practice, we consider two main types of preference datasets: *dense*, where we label comparisons between every sampled segment (effectively ranking all segments), and *sparse*, where we label only one comparison per segment.

**Baseline Methods.** We consider four strong baselines. First, *supervised fine-tuning* (*SFT*) trains a policy with BC on all segments, then fine-tunes on only the preferred segments, i.e., all $\sigma^+$ in $\mathcal{D}_{\text{pref}}$. The second baseline is *Preference IQL* (*P-IQL*), which learns a reward function from $\mathcal{D}_{\text{pref}}$ assuming the partial return model, then subsequently learns a policy to maximize it with Implicit $Q$-Learning (Kostrikov et al., 2022), a state-of-the-art offline RL algorithm. Though P-IQL was first used with the partial return model, here it uses an approximation of $A^*_{r_E}$ as its reward, which as we show in Appendix B's Corollary 1 preserves the optimal policy. In fact, *P-IQL* should be more performant with regret-based labels, since $A^*_{r_E}$ is a highly shaped reward function for $r_E$ Ng et al. (1999); Knox et al. (2023). We use Hejna & Sadigh (2023)'s implementation of P-IQL as it outperformed several contemporary baselines. Third, we consider PPO with a KL-constrained reward as commonly used for RLHF with LLMs (Ziegler et al., 2019). This is *not* a fair comparison, as PPO requires 3.84 million additional online state-action pairs to estimate the policy gradient, totalling $25\times$ the data for CPL 2.5K Dense and $4\times$ the data for CPL 20K Sparse. Unlike the contextual bandit setting used for LLMs, PPO here require full trajectory rollouts. Finally, to demonstrate CPL's ability to extrapolate beyond the best performance in the data, we compare to %BC, where a policy is trained with behavior cloning on the top X% of rollouts according to the ground truth $r_E$.

### 4.1 HOW DOES CPL PERFORM?

**How does CPL perform with state-based observations?** Our main state-based results can be found in rows 1 and 3 of Table 1. When using sparser comparison data (row 3), CPL outperforms prior methods in 5 of 6 environments, often by a substantial margin of over P-IQL, particularly in *Button Press*, *Bin Picking*, and *Sweep Into* environments. When applied to datasets with more dense comparisons, CPL outperforms P-IQL even more (row 1), doing so substantially in all environments. Though the dense-comparison datasets have less state-action coverage, they have substantially more preference comparisons than the sparse comparison datasets. We posit that more comparisons per segment is more beneficial to CPL than to P-IQL because of its contrastive objective – more comparison-rich datasets are likely to have more informative positive-negative pairs that help shape the policy. Moreover, PPO is unable to consistently perform better than CPL despite access to vast amounts of online data from the environment. We found PPO to be very sensitive to the KL-constraint coefficient on reward, which makes it difficult to tune as observed in prior work (Touvron et al., 2023). This problem is likely exacerbated by the instability of the KL-divergence in continuous spaces and the high variance of both value estimation and policy gradients with longer horizons in robotics tasks versus LLM bandits. We find that CPL consistently outperforms %BC, indicating the CPL is indeed exhibiting policy improvement beyond the best behaviors in the dataset.

**How does CPL scale to high-dimensional observations?** To test how CPL's supervised objectives scale to high-dimensional continuous control problems, we render the MetaWorld datasets discussed above to $64 \times 64$ images. We use the network architecture from DrQv2 (Yarats et al., 2022) and the same hyper-parameters as our state-based experiments. We additionally use random shift augmentations, which drastically improve the performance of RL from images (Laskin et al., 2020).

Our image-based results can be found in rows 2 and 4 of Table 1. Interestingly, we find that performance moderately increases for SFT but substantially for P-IQL. We posit that this is because data-augmentation, which is inapplicable in state, plays a key role in improving value representation for P-IQL. Despite this, when learning from denser preference data (row 2), CPL still outperforms P-IQL in 4 of 6 environments and ties on *Sweep Into*. When learning from sparser comparisons (row 4), CPL and P-IQL perform comparably on most tasks, even though CPL is drastically simpler than P-IQL. Again, the gap in performance between CPL and P-IQL is higher with denser comparison data.

These results are only more impressive considering CPL's *significant* reduction in complexity. P-IQL must learn a reward function, a $Q$-function, a value function, and a policy. CPL avoids all of this, and only learns a policy, drastically reducing training time and parameter count. As we can see in Table 2,

| | | Bin Picking | Button Press | Door Open | Drawer Open | Plate Slide | Sweep Into |
|---|---|---|---|---|---|---|---|
| State 2.5k Dense | PPO | $83.7 \pm 3.7$ | $22.7 \pm 1.9$ | $79.3 \pm 1.2$ | $66.7 \pm 8.2$ | $51.5 \pm 3.9$ | $55.3 \pm 6.0$ |
| | SFT | $66.9 \pm 2.1$ | $21.6 \pm 1.6$ | $63.3 \pm 1.9$ | $62.6 \pm 2.4$ | $41.6 \pm 3.5$ | $51.9 \pm 2.1$ |
| | P-IQL | $70.6 \pm 4.1$ | $16.2 \pm 5.4$ | $69.0 \pm 6.2$ | $71.1 \pm 2.3$ | $49.6 \pm 3.4$ | $60.6 \pm 3.6$ |
| | CPL | $\mathbf{80.0 \pm 2.5}$ | $\mathbf{24.5 \pm 2.1}$ | $\mathbf{80.0 \pm 6.8}$ | $\mathbf{83.6 \pm 1.6}$ | $\mathbf{61.1 \pm 3.0}$ | $\mathbf{70.4 \pm 3.0}$ |
| Image 2.5k Dense | SFT | $74.7 \pm 4.8$ | $20.8 \pm 2.4$ | $62.9 \pm 2.3$ | $64.5 \pm 7.6$ | $44.5 \pm 3.2$ | $52.5 \pm 2.5$ |
| | P-IQL | $\mathbf{83.7 \pm 0.4}$ | $22.1 \pm 0.8$ | $68.0 \pm 4.6$ | $76.0 \pm 4.6$ | $51.2 \pm 2.4$ | $\mathbf{67.7 \pm 4.4}$ |
| | CPL | $80.0 \pm 4.9$ | $\mathbf{27.5 \pm 4.2}$ | $\mathbf{73.6 \pm 6.9}$ | $\mathbf{80.3 \pm 1.4}$ | $\mathbf{57.3 \pm 5.9}$ | $\mathbf{68.3 \pm 4.8}$ |
| State 20k Sparse | PPO | $68.0 \pm 4.3$ | $24.5 \pm 0.8$ | $82.8 \pm 1.6$ | $63.2 \pm 6.6$ | $60.7 \pm 4.2$ | $58.2 \pm 0.6$ |
| | SFT | $67.0 \pm 4.9$ | $21.4 \pm 2.7$ | $63.6 \pm 2.4$ | $63.5 \pm 0.9$ | $41.9 \pm 3.1$ | $50.9 \pm 3.2$ |
| | P-IQL | $75.0 \pm 3.3$ | $19.5 \pm 1.8$ | $\mathbf{79.0 \pm 6.6}$ | $76.2 \pm 2.8$ | $\mathbf{55.5 \pm 4.2}$ | $73.4 \pm 4.2$ |
| | CPL | $\mathbf{83.2 \pm 3.5}$ | $\mathbf{29.8 \pm 1.8}$ | $77.9 \pm 9.3$ | $\mathbf{79.1 \pm 5.0}$ | $56.4 \pm 3.9$ | $\mathbf{81.2 \pm 1.6}$ |
| Image 20k Sparse | SFT | $71.5 \pm 1.9$ | $22.3 \pm 2.9$ | $65.2 \pm 2.2$ | $67.5 \pm 1.1$ | $41.3 \pm 2.8$ | $55.8 \pm 2.9$ |
| | P-IQL | $\mathbf{80.0 \pm 2.3}$ | $27.2 \pm 4.1$ | $\mathbf{74.8 \pm 5.8}$ | $\mathbf{80.3 \pm 1.2}$ | $54.8 \pm 5.8$ | $\mathbf{72.5 \pm 2.0}$ |
| | CPL | $78.5 \pm 3.1$ | $\mathbf{31.3 \pm 1.6}$ | $70.2 \pm 2.1$ | $\mathbf{79.5 \pm 1.4}$ | $\mathbf{61.0 \pm 4.2}$ | $72.0 \pm 1.8$ |
| State %BC | 10% | $62.6 \pm 2.6$ | $18.9 \pm 1.7$ | $57.5 \pm 3.0$ | $61.5 \pm 3.7$ | $39.1 \pm 2.5$ | $49.3 \pm 2.1$ |
| | 5% | $64.6 \pm 4.1$ | $18.2 \pm 0.6$ | $59.8 \pm 1.6$ | $61.3 \pm 1.8$ | $38.6 \pm 2.5$ | $49.2 \pm 1.9$ |

Table 1: Success rates ( inpercent) of all methods across six tasks on the MetaWorld benchmark on different datasets. The leftmost column contains the observation modality (state or image), the number of segments in the dataset, and the means of labeling comparisons (dense or sparse). Dense refers to labeling every possible pairwise comparison and sparse refers to labeling only *one* comparison for every *two* segments, e.g., 10k comparisons for 20k segments. We run four seeds for state and three seeds for images. We report the maximum average performance across seeds over an 8-checkpoint, 200 episode evaluation window (details in Appendix E). Bolded values are within 1% of the top-performing method. The bottom section shows oracle performance of %BC with access to ground-truth reward. State-spaces results include PPO (delinated with a dashed line) which is not a 1-1 comparison as it uses 3.84 million extra *online* transitions.

CPL runs $1.62\times$ faster than P-IQL on images and has less than a quarter of the the parameters. As networks get larger and larger, the performance gain from using CPL would only increase.

## 4.2 WHAT CONTRIBUTES TO CPL'S PERFORMANCE?

As alluded to in previous sections, we find that the *gap* in performance between CPL and baselines is higher for datasets with *denser* comparisons. This is consistent with prior works in contrastive learning (Robinson et al., 2021). To study this effect, evaluate CPL's performance as we increase the number of comparisons sampled per segment over a fixed dataset of 5000 segments. We show results of this for *Drawer Open* with state-based observations on the left of Fig. 2 and include the rest in Appendix D.3 in addition to dense data scaling. Overall, we find that CPL benefits from an increasing number of comparisons per segment in all tasks except *Plate Slide*. P-IQL is less affected, though sometimes performs worse with more comparisons, which we suspect is due to reward under-fitting. This highlights another drawback of P-IQL – due to its higher number of components, it has more hyperparameters and is consequently more sensitive to changes in the dataset. We tuned hyperparameters for all methods with 10K comparisons, then left them the same for scaling experiments.

Finally, we ablate both of CPL's hyperparameters – the temperature value $\alpha$ and bias regularizer $\lambda$ – for *Drawer Open* on the right of Fig. 2. While CPL generally performs well with all values, we find that higher performance could have been attained with further hyper-parameter tuning, particularly for $\lambda$. In the Appendix C we ablate more design decisions, like the choice of conservative regularizer.

## 5 RELATED WORK

Though RLHF has recently surged in popularity, learning policies from human preferences has been a long-studied problem, referred to as preference-based RL (PbRL).

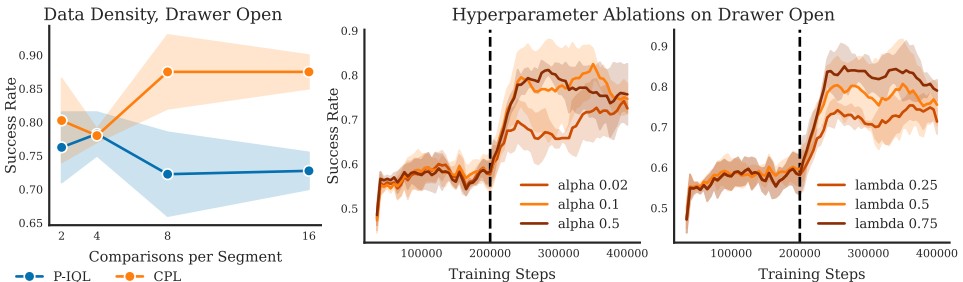

Figure 2: **Left:** Performance when increasing the number of comparisons per segment on Drawer Open state with 5k segments on two seeds. **Right:** Ablations on CPL's hyperparameters on Drawer Open from State. The dotted vertical line shows when BC pretraining stops.

PbRL methods typically start by learning a reward function, usually from pairwise comparisons, then use an RL algorithm for policy optimization (Fürnkranz et al., 2012). While Akrour et al. (2012; 2011); Wilson et al. (2012) were some of the first examples of PbRL, more recently several works have shown that, provided thousands of queries or sufficient pretraining, PbRL can train deep neural-network policies for control using comparisons (Christiano et al., 2017; Lee et al., 2021; Ibarz et al., 2018; Brown et al., 2020; Hejna & Sadigh, 2022; Shin & Brown, 2021) or rankings (Brown et al., 2019; Bıyık et al., 2019; Sikchi et al., 2023a). These approaches, however, are generally demonstrated only on low-dimensional state-based control because of the challenges RL faces when scaling to larger inputs and networks (Ota et al., 2021). In the past, removing RL has lead to effective algorithms for goal-conditioned RL from images (Hejna et al.; Eysenbach et al., 2022). CPL does the same but for PbRL. Other works address the problem of selecting feedback (Sadigh et al., 2017; Biyik et al., 2020; Daniel et al., 2015), which we consider complementary because CPL can benefit from higher quality data elicitation.

To scale RLHF, recent approaches for refining LLMs have ignored the temporal component of RL, and instead treated text-generation as a contextual bandits problem (Ziegler et al., 2019). While this approach has proven effective at tasks like summarization (Stiennon et al., 2020; Wu & Hu, 2018), instruction following (Ouyang et al., 2022; Nakano et al., 2021), and even image generation (Lee et al., 2023; Black et al., 2023), it fundamentally ignores the fact that interaction with users is often sequential,

| Method | Params | Runtime |
|--------|--------|---------|
| P-IQL  | 9.6m   | 16.5 hrs |
| CPL    | 2.1m   | 10.2 hrs |

Table 2: Computational efficiency of each method when learning from pixels for 200k training steps on a single TitanRTX GPU.

spanning multiple turns. Unlike these methods, CPL works with general MDPs. CPL's unique ability to learn from sequence data with only supervised objectives makes it a prime candidate for scaling to more complex problems. In fact, Direct Preference Optimization (DPO) (Rafailov et al., 2023) recently demonstrated that a supervised objective similar to CPL works better than RL in the contextual bandits setting. We show in Appendix B that DPO can be derived as a special case of CPL in which segments are of length 1 and always start at the same state. This parallels Knox et al. (2023), who show that the common contextual bandit-approach is a special case of the naïve approach from Section 3.

To derive CPL's objective, we leverage knowledge from works building on the principle of maximum entropy in control (Ziebart et al., 2008; Ziebart, 2010; Haarnoja et al., 2017). The resulting contrastive update directly learns the optimal policy with fully off-policy data. This is unlike many RL-based RLHF algorithms in both langauge (Ziegler et al., 2019) or control (Christiano et al., 2017) which require on policy rollouts and additional learned components that have been shown to increase variance (Hejna & Sadigh, 2023). Similar contrastive learning objectives have shown to be effective for temporal representation learning (Ma et al., 2023), even with preference data (Kang et al., 2023; Xu et al., 2023).

## 6 DISCUSSION

In this work we introduce CPL, a novel framework for RLHF using the regret preference model. Theoretically, we proved that CPL always learns a *consistent* advantage function and converges to the optimal policy for the expert's reward function. Practically, we showed that CPL's supervised objective is able to outperform RL baselines when learning complex manipulation policies from dense preference data while being simpler and $1.6\times$ faster. Due to space constraints, we include limitations in Appendix A.

ACKNOWLEDGEMENTS

This work was supported by NSF Award 2006388, NSF Award 2218760, NSF Award 1933032, NSF Award 1953032, NSF Award 1941722, NSF Award 2125511, NSF Award IIS-1749204, AFOSR YIP, AFOSR (FA9550-20-1-0077), ARO (78372-CS, W911NF-19-2-0333), ONR (N00014-21-1-2685), ONR, Ford, DARPA YFA, and the Center for AI Safety. JH is supported by a DoD NDSEG Fellowship. CF is a CIFAR Fellow in the Learning in Machines and Brains program. WBK is supported by UT Austin's Good Systems grand challenge. We would like to thank Archit Sharma for valuable discussions on the conservative regularizer used in CPL. Any opinions, findings, and conclusions or recommendations expressed in this material are those of the author(s) and do not necessarily reflect the views of the sponsors.

CONTRIBUTIONS

JH led the project, contributing to all aspects including ideation, theory, experimentation, and writing. RR proposed linking advantages and likelihoods and contributed to early stage ideation. HS contributed to the theory, experiment design, and ran experiments. CF, SN, WBK, DS oversaw, advised, and provided feedback on the project.

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

## A  DISCUSSION

**Limitations.** CPL, like other RLHF approaches, assumes knowledge of the human rater's temporal discounting (i.e., of the discount factor $\gamma$), which in practice would be difficult to communicate. As CPL's loss function is computed over segments, it requires a substantial amount of GPU memory for large segment sizes. Finally, no model of human behavior is perfect.

**Future Directions.** Several exciting research directions remain. First is scaling CPL to larger datasets and architectures where we believe its benefits will be more pronounced. One potentially exciting application is LLMs, where CPL enables fine-tuning on multiple steps of turn-based dialogue. To our knowledge, no multi-step preferences dataset currently exists for LLMs. Second, our work only considers offline data generated by suboptimal policies. An online version of CPL could be developed that works with online human feedback, allowing policies to continually improve.

## B  THEORY

### B.1  PROOF OF CONSISTENCY

We first prove a lemma about the consistency of CPL as it is used when proving convergence.

**Lemma 1.** *Any function $A(s,a)$ that satisfies $\int_{\mathcal{A}} e^{A(s,a)/\alpha} da = 1 \ \forall s \in \mathcal{S}$ is a consistent advantage function under some reward function $r$ in the MaxEntRL setting.*

*Idea.* Given advantage $A(s,a)$, we want to show that there exists a reward function $r$ for which $A$ is the optimal advantage function.

*Proof.* Given $\int_{\mathcal{A}} e^{A(s,a)/\alpha} da = 1$, consider the corresponding policy $\pi^A(a|s) = e^{A(s,a)/\alpha}$. Let the reward function be the advantage, or $r(s,a) = A(s,a) = \alpha \log \pi^A(a|s)$. We can determine the optimal policy $\pi^*$ for this reward according to Eq. (1):

$$\pi^* = \operatorname*{argmax}_{\pi} \mathbb{E}_{\rho_{\pi}}[r(s,a) - \alpha \log \pi(a|s)]$$

$$= \operatorname*{argmax}_{\pi} \sum_{t=1}^{\infty} \mathbb{E}_{s \sim \rho_{\pi}^t(s), a \sim \pi(a|s)}[\alpha \log \pi^A(a|s) - \alpha \log \pi(a|s)]$$

$$= \operatorname*{argmax}_{\pi} \sum_{t=1}^{\infty} \mathbb{E}_{s \sim \rho_{\pi}^t(s)}[-\alpha D_{KL}(\pi(\cdot|s)||\pi^A(\cdot|s))]$$

$$= \operatorname*{argmin}_{\pi} \sum_{t=1}^{\infty} \mathbb{E}_{s \sim \rho_{\pi}^t(s)}[\alpha D_{KL}(\pi(\cdot|s)||\pi^A(\cdot|s))]$$

Thus, the objective is point-wise maximized if and only if $\pi^A(\cdot|s) = \pi(\cdot|s) \ \forall s \in \mathcal{S}$. Therefore, $\pi^A$ is the optimal policy for reward function $r(s,a) = A(s,a)$.[1] Under this reward function, $\pi^* = \pi^A = e^A$, which implies that $A$ is a consistent advantage function.

**Proposition 1.** *CPL learns a consistent advantage function.*

Optimization via CPL fits a valid policy $\pi$ subject to $\int_{\mathcal{A}} \pi(a|s) da = 1 \ \forall s \in \mathcal{S}$, with corresponding MaxEnt Advantage function $A(s,a) = \alpha \log \pi(a|s)$.

$$\int_{\mathcal{A}} e^{A(s,a)/\alpha} da = \int_{\mathcal{A}} e^{\alpha \log \pi(a|s)/\alpha} da = \int_{\mathcal{A}} \pi(a|s) da = 1$$

Thus, by the above Lemma CPL fits a consistent advantage function.

---

[1]Note that we assume that all states are reachable and therefore have support in $\rho_{\pi}^t(s)$ for any optimal MaxEnt policy.

**Corollary 1.** *The reward function $r$ and the reward function defined as the optimal advantage function for $r$, $A_r^*$, have the same optimal MaxEnt policy.*

This corollary can be seen by examining the proof of Lemma 1. According to the MaxEnt RL objective for reward $r$ the optimal policy is $\pi_r^* = e^{A_r^*/\alpha}$ (Ziebart, 2010). Therefore $A_r^* = \alpha\log\pi_r^*$. Repeating the steps of Lemma 1 by setting $r' = A_r^* = \alpha\log\pi_r^*$, we get the following objective for the optimal policy $\pi_{r'}^*$ with respect to $r'$:

$$\pi_{r'}^* = \operatorname*{argmax}_\pi \mathbb{E}_{\rho_\pi}[r'(s,a) - \alpha\log\pi(a|s)]$$

$$= \operatorname*{argmax}_\pi \sum_{t=1}^\infty \mathbb{E}_{s\sim\rho_\pi^t(s),a\sim\pi(a|s)}[\alpha\log\pi_r^*(a|s) - \alpha\log\pi(a|s)]$$

$$= \operatorname*{argmax}_\pi \sum_{t=1}^\infty \mathbb{E}_{s\sim\rho_\pi^t(s)}[-\alpha D_{KL}(\pi(\cdot|s)||\pi_r^*(\cdot|s)]$$

$$= \operatorname*{argmin}_\pi \sum_{t=1}^\infty \mathbb{E}_{s\sim\rho_\pi^t(s)}[\alpha D_{KL}(\pi(\cdot|s)||\pi_r^*(\cdot|s)]$$

Since the final expression above is minimized only when $\pi = \pi_r^*$, then $\pi_{r'}^* = \pi_r^*$. In other words, the reward function $r$ and reward function $r' = A_r^*$ have the same optimal MaxEnt policy.

**Implication for our *P-IQL* baseline.** With regret-based preferences, an algorithm that learns a reward function while assuming the partial return preference model and then conducts RL on that learned reward function—including P-IQL—is actually using an approximation of $A_r^*$ as the reward function. This corollary therefore implies that if that approximation is error-free, then P-IQL is using a reward function that preserves the optimal policy of the expert user's reward function $r_E$. This application of the corollary extends the similar insight of Knox et al. (2023) to the MaxEnt RL setting. Furthermore, as shown in Lemma 1, when $\pi = \pi^*$, $\mathbb{E}_{\rho_\pi}[r(s,a) - \alpha\log\pi(a|s)] = 0$, implying that $V^*(s) = 0$ as the reward and entropy regularization over the occupancy measure from any state is exactly the value function. Thus, as originally pointed out by (Ng et al., 1999), using $A_r^*$ as the reward function results in a high amount of shaping, so much so that a horizon of one transition is sufficient to determine the optimal action in each state (Knox et al., 2023).

### B.2 PROOF OF CONVERGENCE

**Theorem 1.** *Assume an unbounded number of preferences generated from a noisy rational regret-preference model with expert advantage function $A^*$. CPL recovers the optimal policy $\pi^*$.*

*Proof.* Without loss of generality we let $\alpha = 1$. For the purposes of this proof only, let $\sigma_k$ denote a segment of length $k$ where the state-actions in the segment are denoted by $\sigma_k = (s_0,a_0,s_1,a_1,...,s_{k-1},a_{k-1})$. Let $y$ be the label indicating whether the expert regret preference model prefers $\sigma_k^1$ to $\sigma_k^0$, i.e., $y \sim P_{A^*}[\sigma_k^1 \succ \sigma_k^0]$. Let $\hat{A} = \log\hat{\pi}$ be the implicit estimate of $A^*$ learned by CPL. For brevity, we will use the shorthand $A(\sigma_k) = \sum_{\sigma_k}\gamma^t A(s_t,a_t)$ to denote the discounted sum of advantages of a segment $\sigma$. Let $P(\sigma_k^1,\sigma_k^2) = \text{Bern}(\frac{e^{A^*(\sigma^1)}}{e^{A^*(\sigma^1)}+e^{A^*(\sigma^0)}})$ and $Q(\sigma_k^1,\sigma_k^2) = \text{Bern}(\frac{e^{\hat{A}(\sigma^1)}}{e^{\hat{A}(\sigma^1)}+e^{\hat{A}(\sigma^0)}})$ The cross-entropy CPL loss function can be re-written as follows:

$$\mathcal{L}_{\text{CPL}}(\hat{A},\mathcal{D}) = \mathbb{E}_{\sigma^1,\sigma^0\sim\mathcal{D}}[D_{KL}(P(\sigma_k^1,\sigma_k^2)||Q(\sigma_k^1,\sigma_k^2))]$$

The KL divergence is optimized only when the two distributions are exactly equal. Because the preference model is rational and we assume sufficient representation power and unbounded data, it is possible for the loss to converge to zero by pointwise matching KL-divergence for each two comparisons (See Knox et al. (2022) for more information specific to the *identifiability* of regret based preferences). Thus, under the assumption of unbounded data, for all possible segments $\sigma_k^1,\sigma_k^0$ we must have that

$$\frac{e^{A^*(\sigma_k^1)}}{e^{A^*(\sigma_k^1)}+e^{A^*(\sigma_k^0)}} = \frac{e^{\hat{A}(\sigma_k^1)}}{e^{\hat{A}(\sigma_k^1)}+e^{\hat{A}(\sigma_k^0)}} \quad \forall\sigma_k^1,\sigma_k^0.$$

Rearranging, we get:

$$e^{\hat{A}(\sigma_k^1)}e^{A^*(\sigma_k^0)}=e^{A^*(\sigma_k^1)}e^{\hat{A}(\sigma_k^0)}$$

Consider $\sigma_k^p=(s_0^p,a_0^p,s_1^p,a_1^p...s_{k-1}^p,a_{k-1}^p)$ where $p\in\{0,1\}$. We will show that the above equality also holds for all sequences of length $k-1$. Consider the last action for the segment $\sigma_k^1$ denoted as $a_{k-1}^1$, then:

$$\forall\sigma_k^1,\sigma_k^0\sum_{a_{k-1}^1\in\mathcal{A}}e^{\hat{A}(\sigma_k^1)}e^{A^*(\sigma_k^0)}=\sum_{a_{k-1}^1\in\mathcal{A}}e^{A^*(\sigma_k^1)}e^{\hat{A}(\sigma_k^0)}$$

Now, we will use the *consistency* of CPL. Per Ziebart (2010), for the optimal value function $A^*$, $\sum_{a\in\mathcal{A}}e^{A^*(s,a)}=1, \forall s$. Because CPL is *consistent* (Proposition 1), we also have that $\sum_{a\in\mathcal{A}}e^{\hat{A}(s,a)}=1, \forall s$. We use this, in combination with the fact that all possible dynamically feasible segments of length $k-1$ are a subset of dynamically feasible segments of length $k$ to arrive at:

$$\forall\sigma_{k-1}^1,\sigma_k^0,\quad e^{\hat{A}(\sigma_{k-1}^1)}e^{A^*(\sigma_k^0)}=e^{A^*(\sigma_{k-1}^1)}e^{\hat{A}(\sigma_k^0)}$$

Inductively we have that:

$$\forall\sigma_k^0,\quad e^{A^*(\sigma_k^0)}=e^{\hat{A}(\sigma_k^0)}$$

Applying the same argument again, this time for $\sigma_k^0$, we have

$$\forall s_i^0,a_i^0\ \ e^{A^*(s_i^0,a_i^0)}=e^{\hat{A}(s_i^0,a_i^0)}$$

which is equivalent to $A^*(s,a)=\hat{A}(s,a)\ \forall s,a$.

### B.3 CONVEXITY OF CPL WITH FINITE DATA

**CPL is convex, but not *strictly* convex.** Here we show that the CPL loss function is convex in $\log\pi$. Consider the logistic regression interpretation of CPL for finite data

$$\mathcal{L}_{\mathrm{CPL}}(\pi,\mathcal{D}_{\mathrm{pref}})=-\sum_{i=1}^{n}\log\mathrm{logistic}(\alpha x_i^\top\log\pi(a|s)),$$

where $x_i$ is the "comaprison" vector for the $i$th comparison in $\mathcal{D}_{\mathrm{pref}}$. We can re-write this using matrix notation as:

$$\mathcal{L}_{\mathrm{CPL}}(\pi,\mathcal{D}_{\mathrm{pref}})=-\sum_{i=1}^{n}\log\mathrm{logistic}((\alpha X\log\pi(a|s))_i).$$

The hessian of this objective (logistic regression) with respect to $\log\pi$ is $X^\top DX$, where $D$ is the diagonal matrix such that $D_{ii}=\mathrm{logistic}(x_i\cdot\log\pi)(1-\mathrm{logistic}(x_i\cdot\log\pi))$. As $X^\top DX$ is symmetric, it is guaranteed to be positive semi-definite making the objective function convex. The distributional constraint of CPL, that $\forall s\in\mathcal{S},\int_{\mathcal{A}}e^{\log\pi(a|s)}da=1$, is also convex as $e^{\log\pi}$ is convex in $\log\pi$. Thus, the overall objective is convex.

However, this does not imply strict convexity, or that there is a unique solution. $X^\top DX$ is only positive definite if it is full rank, which is unlikely to happen in practice as usually $|\mathcal{S}\times\mathcal{A}|>>n$. This means that the objective is likely not *strictly* convex in practice, as there can exist more than one minimizer of the objective, formally denoted $\hat{\pi}=\mathrm{argmin}_\pi\mathcal{L}_{\mathrm{CPL}}(\pi,\mathcal{D}_{\mathrm{pref}})$. To prove that CPL is not always strictly convex, we construct another policy $\hat{\pi}'$ such that $\mathcal{L}_{\mathrm{CPL}}(\hat{\pi},\mathcal{D}_{\mathrm{pref}})=\mathcal{L}_{\mathrm{CPL}}(\hat{\pi}',\mathcal{D}_{\mathrm{pref}})$. First, we demonstrate this on a simple single-state MDP and then provide a general construction for arbitrary MDPs with discrete actions.

**A simple example.** Consider a single state MDP with three actions $a^1,a^2,a^3$ and expert reward function $r_E(s,a^i)=r^i$ where $i$ indexes the actions. It can be shown that, due to the single state nature of this simple MDP, the optimal maximum entropy advantage function is $A^*(s,a^i)=r^i$. Consider a preference dataset $\mathcal{D}_{\mathrm{pref}}$ consisting only of comparisons between segments $(s,a^1)$ and

$(s,a^2)$. According to the regret preference model, the expert labels these preferences according to $\text{Bern}\big(\exp r^1/(\exp r^1+\exp r^2)\big)$ and thus we expect some labels in the preference matrix $X$ to conflict. The finite CPL loss becomes

$$\mathcal{L}_{\text{CPL}}(\pi,\mathcal{D}_{\text{pref}})=-c_1\log\text{logistic}\big(\alpha\log\pi(a^1|s)-\alpha\log\pi(a^2|s)\big)$$
$$-c_2\log\text{logistic}\big(\alpha\log\pi(a^2|s)-\alpha\log\pi(a^1|s)\big)$$

where $c_1$ and $c_2$ are the number of comparisons where $a^1$ and $a^2$ were preferred respectively. By taking the gradient of this objective, it can be shown that the loss is optimized only when $\text{logistic}\big(\alpha\log\pi(a^1|s)-\alpha\log\pi(a^2|s)\big)=\frac{c_1}{c_1+c_2}$ or reducing, $\alpha\log\pi(a^1|s)-\alpha\log\pi(a^2|s)=\log\frac{c_1}{c_2}$. Intuitively, this makes sense, as the logits are optimized to produce the same ratio of preferences as found in the dataset. However, when we consider the unseen action $a^3$, to which we can assign arbitrary probability, the existence of multiple optimizers $\hat{\pi}$ becomes clear. For example, take $c_1=c_2$. By the conditions above its straightforward to see that $\hat{\pi}=[0.5,0.5,0.0]$ is an optimum of the CPL loss function. However, $\hat{\pi}=[0.1,0.1,0.8]$ achieves the same loss as its difference in log probabilities $\log\pi(a^1|s)-\log\pi(a^2|s)$ is the same. If $c_2=0$, or we have no conflicting preferences, $\hat{\pi}=[1,0,0]$ and the implied $\hat{A}=\log\hat{\pi}$ is undefined, implying some of the reward values are infinite. This means we do not have enough data to accurately fit $\pi^*$. Next, we provide a construction for more general MDPs in the presence of OOD actions.

**A more general construction.** Let the expert reward function $r_E$ to be bounded. We will interpret the finite preference dataset as a matrix $X$ as described in Section 3.2 and use $N(X)$ to denote the null space of $X$. For a vector $u\in\mathbb{R}^{|\mathcal{S}\times\mathcal{A}|}$ we use $u(s,a)$ to index the vector at state $s$ and action $a$. Assume the following about $X$:

1. There exists a vector $u\in N(X)$ such that for state action $s,a$ contained in $X$, $u(s,a)\neq 0$. In other words, the null space is non-trival on the support of $\mathcal{D}_{\text{pref}}$.

2. For every state in the dataset where there is an action such that $u(s,a)\neq 0$, there exists at least one out-of-distribution (OOD) action $a_{\text{OOD}}$ *not* in the dataset. The indicator vector for $s,a_{\text{OOD}}$ is thus a basis vector for $N(X)$.

Let $\hat{\pi}$ be the minima of the CPL loss function. We will construct $\hat{\pi}'$ as follows. Select a vector $u\in N(X)$ that is non-zero for at least one $s,a$ pair in $X$. As $u\in N(X)$, we have that $\mathcal{L}_{\text{CPL}}(\hat{\pi},\mathcal{D}_{\text{pref}})=\mathcal{L}_{\text{CPL}}(e^{\log\hat{\pi}+u},\mathcal{D}_{\text{pref}})$. However, $e^{\log\hat{\pi}+u}$ violates the policy constraint as it may not integrate to one. We can fix this problem by adding or removing probability mass from the OOD actions we have assumed exist at states where $u$ is non-zero. We do this by constructing another vector $v\in N(X)$ by choosing one $a_{\text{OOD}}$ at each state without loss of generality. By examining the total sum of probabilities of the modified policy,

$$\forall s\in X,\sum_a\hat{\pi}e^{u(s,a)}=\sum_{a\neq a_{\text{OOD}}}\hat{\pi}(a|s)e^{u(s,a)}+\hat{\pi}(a_{\text{OOD}}|s)$$

we can normalize the sum using the indicator vectors for $s,a_{\text{OOD}}$, which are necessarily in the nullspace $N(X)$. Consider a vector $v$ such that at each state $s$, $v(s,a)=0$ except for at $a_{\text{OOD}}$, where $v(s,a_{\text{OOD}})=\log(1-\sum_{a\neq a_{\text{OOD}}}\hat{\pi}(a|s)e^{u(s,a)})-\log\hat{\pi}(a_{\text{OOD}}|s)$. Then,

$$\forall s\in X,\sum_a\hat{\pi}e^{u(s,a)+v(s,a)}=\sum_{a\neq a_{\text{OOD}}}\hat{\pi}(a|s)e^{u(s,a)}+\hat{\pi}(a_{\text{OOD}}|s)e^{v(s,a_{\text{OOD}})}=1$$

As $v$ is formed from a linear combination of basis vectors of $N(X)$, $v\in N(X)$. Consequently, $\mathcal{L}_{\text{CPL}}(\hat{\pi},\mathcal{D}_{\text{pref}})=\mathcal{L}_{\text{CPL}}(e^{\log\hat{\pi}+u+v},\mathcal{D}_{\text{pref}})$ and by the above construction $\hat{\pi}'=\hat{\pi}e^{u+v}$ is a valid policy. This completes the construction.

We have shown that an infinite number of policies can attain the same optima, just by shifting the amount of probability assigned to OOD actions. For some of these solutions, the entire mode of the policy is potentially out-of-distribution. In the offline setting, the pessimism principle dictates that we should discourage modes that are out-of-distribution. We fix this by introducing regularization.

### B.4 CONSERVATIVE BIAS REGULARIZATION

CPL loss translates a relative weighting between preferences to a policy, but does not employ any mechanism to ensure the learned policy is close to the dataset. In the offline setting, this can be detrimental if the learned policy incorrectly extrapolates to out of distribution actions. A similar approach, under the name of pessimism or conservatism, is commonly seen in offline RL literature (Levine et al., 2020; Jin et al., 2021; Sikchi et al., 2023b). As expalined in Section 3.2, we want to learn policies that have a high-probability on the dataset. However, there are many datasets that potentially have the the *same* loss, as $\mathcal{L}_{\text{CPL}}$ depends only on the *difference* in probability for each preference comparison, or $\sum_{\sigma^+}\gamma^t\log\pi(a_t|s_t)-\sum_{\sigma^-}\gamma^t\log\pi(a_t|s_t)$, and thus constants added to the log probabilities of each segment cancel. However, we would prefer that a higher loss is given when the policy is assigns lower probability to actions in the dataset.

To remedy this, we introduced bias regularizer $\lambda \in (0,1)$ in Section 3, which leads to the modified preference loss:

$$\mathcal{L}_{\text{CPL}(\lambda)}(\pi,\mathcal{D}_{\text{pref}})=\mathbb{E}_{\mathcal{D}_{\text{pref}}}\left[-\log\frac{\exp\sum_{\sigma^+}\gamma^t\alpha\log\pi(a_t^+|s_t^+)}{\exp\sum_{\sigma^+}\gamma^t\alpha\log\pi(a_t^+|s_t^+)+\exp\lambda\sum_{\sigma^-}\gamma^t\alpha\log\pi(a_t^-|s_t^-)}\right].$$

Next, we prove that this loss discourages the policy from learning modes that are out-of-distribution, starting with the proposition from the main text.

**Proposition 2.** *Consider a comparison $\sigma^+ \succ \sigma^-$ from $\mathcal{D}_{pref}$ and an arbitrary comparison $\sigma'^+ \succ \sigma'^-$ such that $\mathcal{L}_{CPL}(\pi,\sigma^+ \succ \sigma^-)=\mathcal{L}_{CPL}(\pi,\sigma'^+ \succ \sigma'^-)$ for a fixed policy $\pi$. If $\sum_{\sigma^+}\gamma^t\log\pi(a_t^+|s_t^+) > \sum_{\sigma'^+}\gamma^t\log\pi(a_t^+|s_t^+)$, then $\mathcal{L}_{CPL(\lambda)}(\pi,\sigma^+ \succ \sigma^-) < \mathcal{L}_{CPL(\lambda)}(\pi,\sigma'^+ \succ \sigma'^-)$.*

Succinctly, this proposition states that if preference comparisons each achieve the same loss, the less likely comparisons under the policy (in this case $\sigma'^+ \succ \sigma'^-$), will have higher regularized CPL loss. Essentially, this shows that the regularized objective encourages the policy to have higher likelihood on the provided comparisons than any other potential comparison that exists. *Proof.* By the stated assumptions, it must be that $\sum_{\sigma'^+}\gamma^t\log\pi(a_t|s_t)+\delta=\sum_{\sigma^+}\gamma^t\log\pi(a_t|s_t)$ for some $\delta>0$. As the two comparisons also have the same CPL Loss, their logits must be the same, or

$$\sum_{\sigma^+}\gamma^t\log\pi(a_t|s_t)-\sum_{\sigma^-}\gamma^t\log\pi(a_t|s_t)=\sum_{\sigma'^+}\gamma^t\log\pi(a_t|s_t)-\sum_{\sigma'^-}\gamma^t\log\pi(a_t|s_t).$$

Consequently, the same $\delta$ must hold for the negative segments, or $\sum_{\sigma'^-}\gamma^t\log\pi(a_t|s_t)+\delta=\sum_{\sigma^-}\gamma^t\log\pi(a_t|s_t)$. We can then examine the regularized CPL loss under each comparison. First, we evaluate the finite regularized loss for $\sigma^+ \succ \sigma^-$, algebraically simplified for clarity:

$$\mathcal{L}_{\text{CPL}(\lambda)}(\pi,\sigma^+ \succ \sigma^-)=-\log\frac{\exp\sum_{\sigma^+}\gamma^t\alpha\log\pi(a_t|s_t)}{\exp\sum_{\sigma^+}\gamma^t\alpha\log\pi(a_t|s_t)+\exp\lambda\sum_{\sigma^-}\gamma^t\alpha\log\pi(a_t|s_t)}$$

$$=-\log\text{logistic}\left(\sum_{\sigma^+}\gamma^t\log\pi(a_t|s_t)-\lambda\sum_{\sigma^-}\gamma^t\log\pi(a_t|s_t)\right)$$

$$=\log\left(1+\exp\left(\lambda\sum_{\sigma^-}\gamma^t\log\pi(a_t|s_t)-\sum_{\sigma^+}\gamma^t\log\pi(a_t|s_t)\right)\right)$$

We can then compare this to the regularized loss for $\sigma'^+ \succ \sigma'^-$.

$$\mathcal{L}_{\text{CPL}(\lambda)}(\pi,\sigma'^+ \succ \sigma'^-)=\log\left(1+\exp\left(\lambda\sum_{\sigma'^-}\gamma^t\log\pi(a_t|s_t)-\sum_{\sigma'^+}\gamma^t\log\pi(a_t|s_t)\right)\right)$$

$$=\log\left(1+\exp\left(\lambda\sum_{\sigma^-}(\gamma^t\log\pi(a_t|s_t)-\delta)-\sum_{\sigma^+}(\gamma^t\log\pi(a_t|s_t)-\delta)\right)\right)$$

$$=\log\left(1+\exp(\delta(1-\lambda))\exp\left(\lambda\sum_{\sigma^-}\gamma^t\log\pi(a_t|s_t)-\sum_{\sigma^+}\gamma^t\log\pi(a_t|s_t)\right)\right)$$

The key step in the above is substituting the relationship between the log probabilities of the comparisons. As $\delta > 0$ and $0 < \lambda < 1$, it can easily be seen that the loss is lower for $\sigma^+ \succ \sigma^-$, letting us conclude that

$$\mathcal{L}_{\mathrm{CPL}(\lambda)}(\pi, \sigma^+ \succ \sigma^-) < \mathcal{L}_{\mathrm{CPL}(\lambda)}(\pi, \sigma'^+ \succ \sigma'^-)$$

We can extend this proposition to the regularized CPL loss over entire datasets as follows:

*For a fixed policy $\pi$, consider two preference datasets $\mathcal{D}_n = \{(\sigma_i^+, \sigma_i^-)\}_{i=1}^n$ and $\mathcal{D}'_n = \{(\sigma_i'^+, \sigma_i'^-)\}_{i=1}^n$ such that $\forall m = 1, 2, ..., n, \mathcal{L}_{CPL}(\mathcal{D}_m, \pi) = \mathcal{L}_{CPL}(\mathcal{D}'_m, \pi)$. Then, if $\sum_{\sigma_i'^+} \gamma^t \log \pi(a_t|s_t) \leq \sum_{\sigma_i^+} \gamma^t \log \pi(a_t|s_t)$ for all $i$ and strictly for at least one $i$,*

$$\mathcal{L}_{CPL(\lambda)}(\mathcal{D}_n, \pi) < \mathcal{L}_{CPL(\lambda)}(\mathcal{D}'_n, \pi)$$

The proof of this amounts to first noticing that, because the preference losses are the same for every ordered subset, the losses for the $i$th datapoints in $\mathcal{D}'_n$ and $\mathcal{D}_n$ must be the same. Then, we can repeatedly apply Proposition 2. Since the inequality is strict at at-least one datapoint, the regularized loss will be strictly lower.

We can construct datasets for which this is applicable. For example, consider a dataset $\mathcal{D}$ containing a total ordering over $n$ segments, $\sigma^1 \succeq \sigma^2 \succeq ... \succeq \sigma^n$. The unregularized loss for this policy and dataset is $\mathcal{L}_{\mathrm{CPL}}(\pi, \mathcal{D})$. We can construct another dataset $\mathcal{D}'$ over a *different* set of totally ordered segments from anywhere in the state space $\sigma'^1 \succeq \sigma'^2 \succeq .. \succeq \sigma'^n$ such that:

$$\sum_{\sigma'^i} \gamma^t \log \pi(a_t|s_t) + \delta = \sum_{\sigma^i} \gamma^t \log \pi(a_t|s_t)$$

for all $i = 1, 2, ..., n$ and some $\delta \geq 0$.

## B.5    CPL FOR RANKINGS

We can derive a version of CPL for ranking data using a Plackett-Luce model (Plackett, 1975). We denote the chosen ranking as a permutation $\tau : [K] \to [K]$ where $K$ is the number of segments presented, $\sigma^1, ..., \sigma^K$. The Plackett-Luce model under regret based preferences is:

$$P(\tau|\sigma^1, ..., \sigma^K) = \prod_{k=1}^K \frac{\exp \sum_{\sigma^{\tau(k)}} \gamma^t A^*(s_t^{\tau(k)}, a_t^{\tau(k)})}{\sum_{j=k}^K \exp \sum_{\sigma^{\tau(j)}} \gamma^t A^*(s_t^{\tau(j)}, a_t^{\tau(j)})}$$

This model generalizes to Bradley-Terry (Bradley & Terry, 1952) when $K = 2$. To learn the optimal policy, we maximize the log likelihood of the above and make the same substitution as CPL, $\alpha \log \pi^*(a|s) = A^*(s, a)$. This gives us the CPL loss function for rankings, which can be seen as a verison of the InfoNCE objective. Without loss of generality, we order the permutations $\tau$ such that $\sigma^1 \succeq \sigma^2 \succeq ... \succeq \sigma^K$.

$$\mathcal{L}_{\mathrm{CPL}}(\pi_\theta, \mathcal{D}_{\mathrm{rank}}) = \mathbb{E}_{(\sigma^1, ..., \sigma^K) \sim \mathcal{D}_{\mathrm{rank}}} \left[ -\sum_{k=1}^K \log \frac{\exp \sum_{\sigma^{\tau(k)}} \gamma^t \alpha \log \pi_\theta(a_t^k|s_t^k)}{\sum_{j=k}^K \exp \sum_{\sigma^{\tau(j)}} \gamma^t \alpha \log \pi_\theta(a_t^j|s_t^j)} \right]$$

Except for the sum over $k$, this is the exact objective from Oord et al. (2018) where the scores are the discounted sum of log probabilities over the segments.

## B.6    DIRECT PREFERENCE OPTIMIZATION AS SPECIAL CASE OF CPL

**Reduction via Maximum Entropy Advantage.** Note that by the Bellman equation,

$$A^*(s, a) = Q^*(s, a) - V^*(s, a) = r_E(s, a) + \gamma \mathbb{E}_{s'}[V^*(s')] - V^*(s)$$

DPO (Rafailov et al., 2023) assumes the *contextual-bandits* setting, thus the MDP terminates after a single step and there is no next state $s'$. As we can see from the above, in this setting, $A^*(s, a) = r_E(s, a) - V^*(s)$. DPO also assumes that all preferences start from the same state $s$, and thus only actions $a^+$ and $a^-$ differ. This is consistent with RLHF on LLMs as humans score "responses" to fixed prompts.

The regret preference model becomes:

$$P_{A^*}\left[\sigma^+ \succ \sigma^-\right] = \frac{\exp r_E(s,a^+) - V^*(s)}{\exp r_E(s,a^+) - V^*(s) + \exp r_E(s,a^-) - V^*(s)}$$

$$= \frac{\exp r_E(s,a^+)}{\exp r_E(s,a^+) + \exp r_E(s,a^-)}$$

which is the same preference model used in DPO. From here the same conservative derivation as DPO can be applied by noting that, for KL-constrained contextual bandits, $\pi^*(a|s) = \mu(a|s)e^{Q^*(s,a)-V^*(s)} = \mu(a|s)e^{r_E(s,a)-V^*(s)}$ for reference distribution $\mu$. Solving for $r_E$, we can perform a substitution just like in CPL to arrive at the DPO objective.

**CPL under Constrained Regret Preferences**. We can also consider a setting where users provide preferences constrained to a reference distribution $\mu$. This might arise in scenarios where users are only shown a fixed set of behaviors, and do not extrapolate far beyond them. Though we do not believe this premise has previously been considered, it leads to an interesting result.

Assume preferences to be distributed according to the $KL$ constrained advantage function. In this setting, $\pi^*(a|s) = \mu(a|s)e^{A^*(s,a)}$ and by substitution the CPL loss becomes

$$\mathcal{L}_{\text{CPL}}(\pi_\theta, \mathcal{D}_{\text{pref}}) = \mathbb{E}_{(\sigma^+, \sigma^-) \sim \mathcal{D}_{\text{pref}}}\left[-\log \frac{\exp\sum_{\sigma^+}\gamma^t\alpha\log\frac{\pi_\theta(a_t^+|s_t^+)}{\mu(a_t^+|s_t^+)}}{\exp\sum_{\sigma^+}\gamma^t\alpha\log\frac{\pi_\theta(a_t^+|s_t^+)}{\mu(a_t^+|s_t^+)} + \exp\sum_{\sigma^-}\gamma^t\alpha\log\frac{\pi_\theta(a_t^-|s_t^-)}{\mu(a_t^-|s_t^-)}}\right].$$

which is essentially a multi-step generalization of DPO which has not previously been considered. In the next section, we expand on this as a variant of CPL.

## C  VARIANTS OF CPL

In the main body of the paper, we presented the version of CPL which we found to consistenly attain good performance. In some of our experiments, we also considered two other variants of CPL. We detail these below.

**BC-Regularized CPL.** Instead of using our biased conservative regularization from An et al. (2023), we consider using a simple BC regularizer. This can be derived by considering the objective:

$$\min_\pi \mathcal{L}_{\text{CPL}}(\pi_\theta, \mathcal{D}_{\text{pref}}) \text{ s.t. } \mathbb{E}_{s \sim \rho_\mu}[D_{KL}(\mu(\cdot|s)||\pi(\cdot|s)] < \epsilon$$

Relaxing the problem via Lagrangian duality with langrangian $\beta$, we arrive at a BC regularized version of CPL.

$$\min_\pi \mathcal{L}_{\text{CPL}}(\pi_\theta, \mathcal{D}_{\text{pref}}) - \beta\mathbb{E}_{(a,s) \sim \rho_\mu}[\log\pi(a|s)]$$

We refer to this method as CPL (BC).

**KL Constrained CPL.** We can also consider the setting where preferences are assumed to be distributed according to the constrained advantage function. Though in practice we sample preferences according to the maximum entropy advantage function, we found this approach to still work in many settings. First, we learn the reference distribution $\mu$ using behavior cloning. Then, we use constrained CPL with bias regularization, making the final loss function:

$$\mathcal{L}_{\text{CPL-KL}(\lambda)}(\pi_\theta, \mathcal{D}_{\text{pref}}) = \mathbb{E}_{(\sigma^+, \sigma^-) \sim \mathcal{D}_{\text{pref}}}\left[-\log \frac{\exp\sum_{\sigma^+}\gamma^t\alpha\log\frac{\pi_\theta(a_t^+|s_t^+)}{\mu(a_t^+|s_t^+)}}{\exp\sum_{\sigma^+}\gamma^t\alpha\log\frac{\pi_\theta(a_t^+|s_t^+)}{\mu(a_t^+|s_t^+)} + \exp\sum_{\sigma^-}\lambda\gamma^t\alpha\log\frac{\pi_\theta(a_t^-|s_t^-)}{\mu(a_t^-|s_t^-)}}\right]$$

We refer to this method as CPL (KL).

**CPL with Dense Preferences.** When learning from "dense" preference data, it is possible to augment the batch to include more comparisons using the transitive property. Specifically, given a batch of $b$

segments, we compute all possible pairwise comparisons within a batch:

$$\mathcal{L}_{\text{CPL}(\lambda)\text{-D}} = -\sum_{i=0}^{b-1}\sum_{j=0}^{b-1} 1\{\sigma_i \succ \sigma_j\} \log \frac{\exp\sum_{\sigma^i}\gamma^t\alpha\log\pi_\theta(a_t^i|s_t^i)}{\exp\sum_{\sigma^i}\gamma^t\alpha\log\pi_\theta(a_t^i|s_t^i) + \exp\lambda\sum_{\sigma^j}\gamma^t\alpha\log\pi_\theta(a_t^j|s_t^j)}$$

This provides as much contrastive signal per-batch as possible. We applied this technique to our CPL experiments with images, and found that it lead to a slight increase in performance for some tasks.

## D    EXTENDED RESULTS

In this section we provide our full experimental results:

1. Learning curves from state for CPL, baselines, and variants described in Appendix C.

2. Learning curves from images for CPL and baselines.

3. Scaling results for CPL and P-IQL with different sized dense datasets and fixed sparse datasets with a varying number of comparisons.

4. Results when varying the number of comparisons for a fixed dataset.

5. More Hyper-parameter ablations, include temperature and pretraining.

6. D4RL real human results.

### D.1    STATE LEARNING CURVES

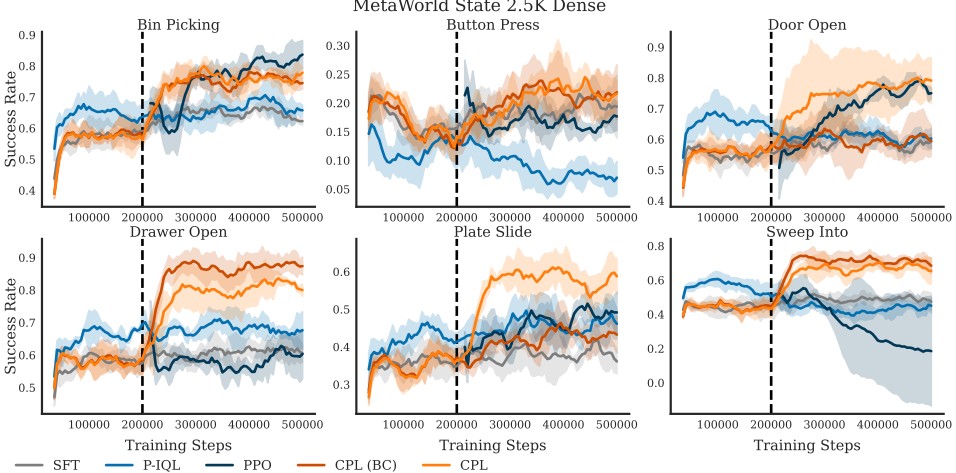

Figure 3: State-based results in MetaWorld with 2.5K segments and dense comparisons. This plot also shows CPL BC. The dotted vertical line indicates when BC pretraining stops for CPL, SFT, and PPO. We show a KL-coefficient of 2 for PPO, which uses 3.84 million extra state-action pairs.

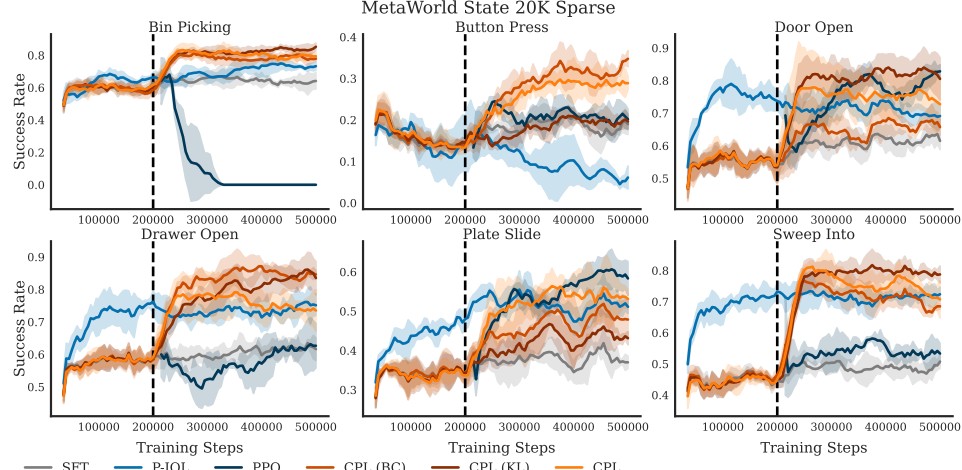

Figure 4: State-based results in MetaWorld with 20K segments and sparse comparisons. This plot also shows CPL variants. The dotted vertical line is when BC pretraining stops for CPL, SFT, and PPO. We show a KL-coefficient of 2 for PPO, which uses 3.84 million extra state-action pairs.

### D.2 IMAGE LEARNING CURVES

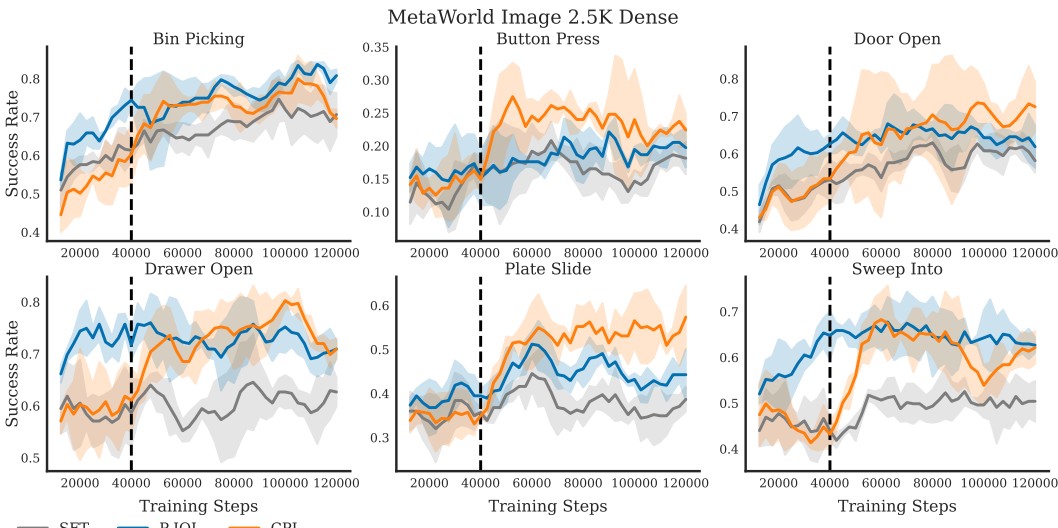

Figure 5: Image-based results in MetaWorld with 2.5K segments and dense comparisons. This plot additionally shows the CPL BC variant. The dotted vertical line indicates when BC pretraining stops for CPL and SFT.

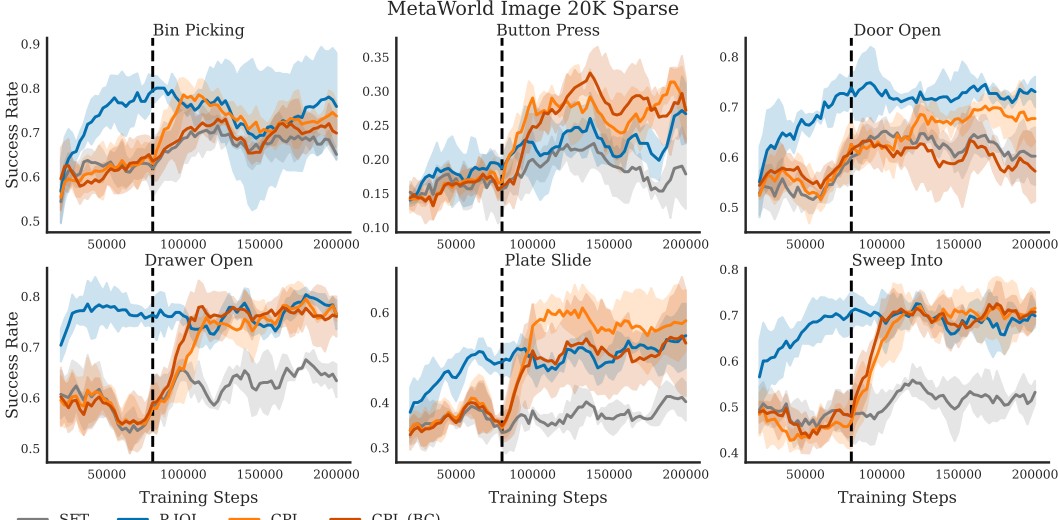

Figure 6: Image-based results in MetaWorld with 20K segments and sparse comparisons. This plot shows CPL variants in addition to baselines. The dotted vertical line indicates when BC pretraining stops for CPL and SFT.

## D.3 DATA SCALING

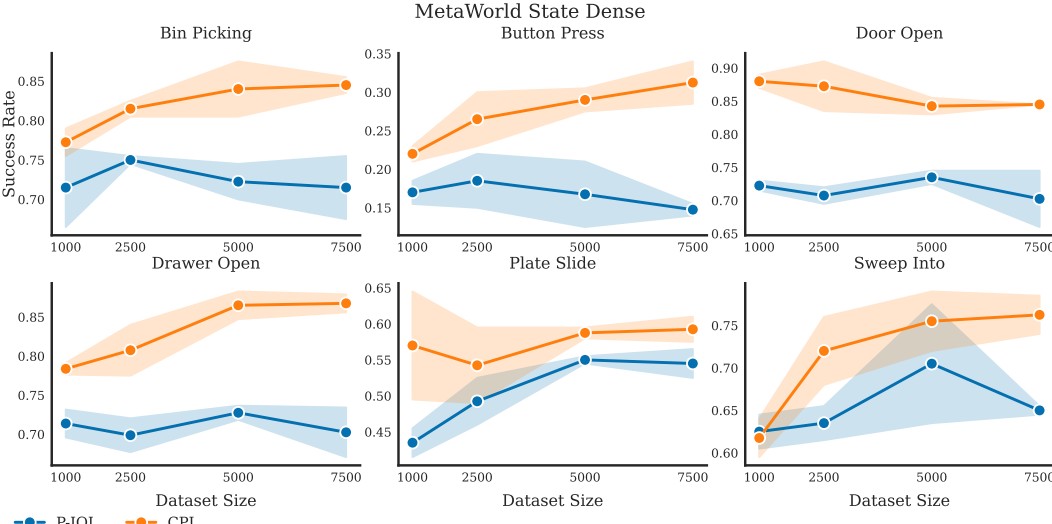

Figure 7: Scaling on state-based MetaWorld environments for different sized dense comparison datasets.

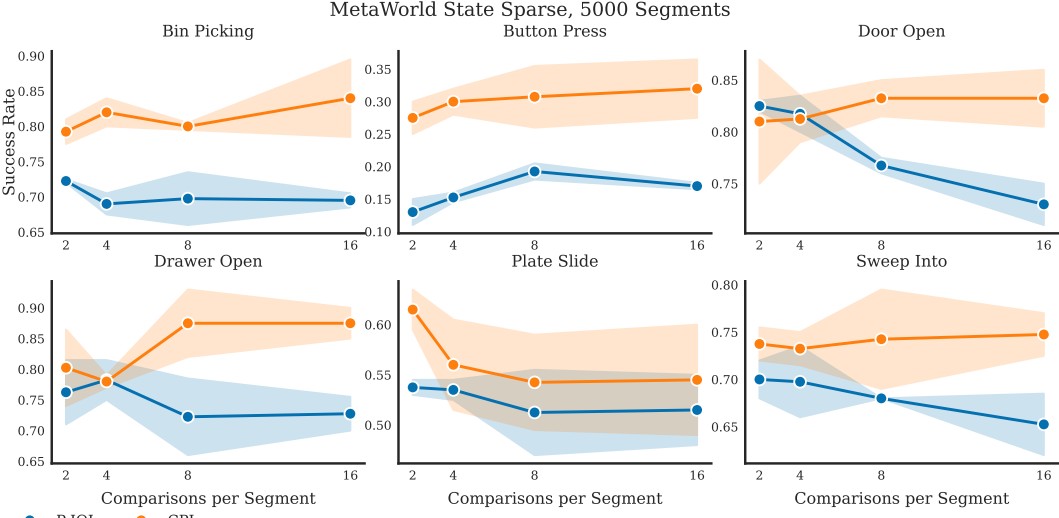

Figure 8: Scaling on state-based MetaWorld environments for 5000 segments varying the number of comparisons per segment. We find that for these results, P-IQL's performance sometimes goes down. Inspecting our training logs reveals that this is likely due to the reward function underfitting. For example, on *Door Open*, the reward modeling loss is near zero with only 2 comparisons per segment (10K comparisons total, which is the amount we tuned our hyper-parameters for). With 16 comparisons per segment, the loss ends near 0.16. This highlights an additional limitation of P-IQL: it requires tuning an addition reward model, which can be very sensitive to its training parameters. CPL removes this additional complexity, and is thus easier to scale.

## D.4 ADDITIONAL ABLATIONS

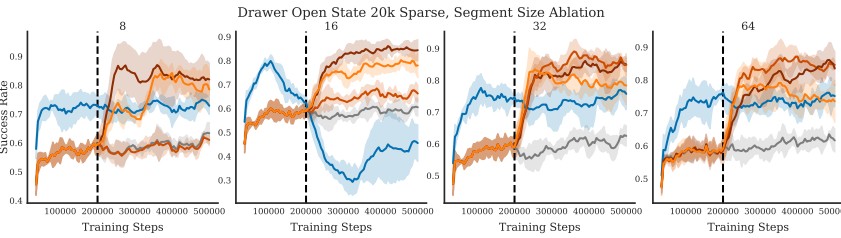

Figure 9: Results varying the size of segments on Drawer Open from State. The dataset was fixed to 20K segments with 10K comparisons, but the size of each segment was varied.

| | Walker-Med-Exp | Walker-Med-Replay | Hopper-Med-Exp | Hopper-Med-Replay |
|---|---|---|---|---|
| P-IQL | $99.9 \pm 6.2$ | $71.6 \pm 5.9$ | $88.6 \pm 3.6$ | $60.2 \pm 20.6$ |
| PT | $\mathbf{110.2} \pm \mathbf{0.8}$ | $\mathbf{76.6} \pm \mathbf{3.2}$ | $86.7 \pm 6.8$ | $\mathbf{78.9} \pm \mathbf{10.3}$ |
| CPL | $\mathbf{109.2} \pm \mathbf{0.2}$ | $48.3 \pm 3.7$ | $\mathbf{109.1} \pm \mathbf{0.7}$ | $72.4 \pm 3.1$ |

Table 3: Results on the D4RL benchmark from real human feedback from Kim et al. (2023). "PT" denotes the original Preference Transformer baseline which uses the IQL algorithm after learning a transformer-based reward model.

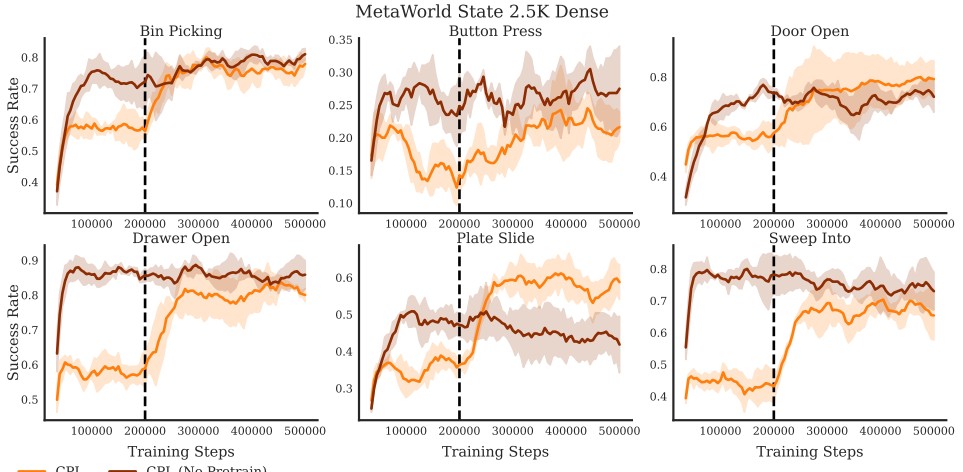

Figure 10: Results with and without pretraining on MetaWorld state with 2.5K Dense segments. Interestingly, performance can be higher without pretraining.

## D.5 D4RL REAL-HUAMN PREFERENCE RESULTS

To demonstrate both the applicability of the regret preference model and performance of CPL, we perform additional experiments with *real-human* preferences. Specifically, we adopt the benchmarks from Kim et al. (2023) which use either 100 (expert) or 500 (replay) real human preferences on datasets from the D4RL benchmark (Fu et al., 2020). To faciliate learning from such a limited number of queries, we modified CPL by first learning a logistic regression model to predict the users preference $P[\sigma^+ \succ \sigma^-]$ following An et al. (2023). Critically, this is not a reward or advantage function as it takes *entire* segments as inputs, not single state-action pairs. We then use this model to relabel the offline D4RL data with dense preferences for CPL. We borrow logistic regression model borrows the architecture from Kim et al. (2023) and do not use BC pretraining. Our results can be found in Table 3. CPL has the best performance by a large margin in Hopper-Medium-Exp, but performs worse in Walker-Medium-Replay. We posit that this is because the preferences for this dataset, may not closely follow the regret-based model as per discussions with the authors of Kim et al. (2023), they were collected by a single user with a pre-planned rules-based approach. Nonetheless, CPL is still able to perform well on these tasks with only 100 or 500 human preferences, and no value or $Q$-function learning.

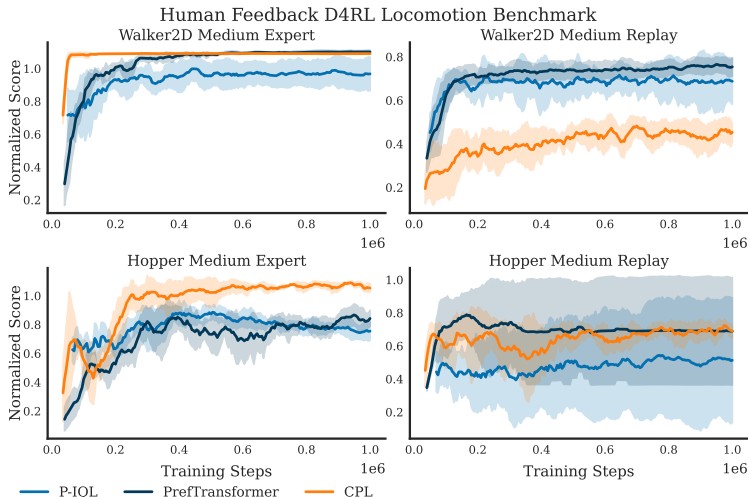

Figure 11: Results for the D4RL human preference datasets.

# E EXPERIMENT DETAILS.

Our datasets and code are publicly released at https://github.com/jhejna/cpl.

## E.1 ENVIRONMENT DETAILS

We use a modified version of the MetaWorld environments (Yu et al., 2020) in our experiments, which we found necessary to obtain good regret-based preference labels. MetaWorld was designed for Meta-RL, and thus by default hides the goal from the state spaces. Prior works like Lee et al. (2021), have randomized the goal but left it hidden, making the reward function stochastic. We randomize the goal, but make it observable to remove reward stochasticity. We additionally randomize the initial position of the arm, which is not done by default. This increases data coverage but also leads to more robust policies. Finally, in MetaWorld v2 the state by default includes object and proprioceptive history. We remove proprioceptive history to make the environment more Markovian.

## E.2 DATASETS AND PREFERENCE LABELING

In Section 4 we provided details on how we generated our datasets. Though we tried to select suboptimal SAC checkpoints that achieves approximately a 50% success rate, there was some variance. In Table 4 we show the overall success rate of trajectories in the rollout dataset for each environment. We also apply gaussian noise of standard deviation 0.3 when collecting rollouts. Next, we provide further details on how we generated accurate regret-based labels.

| Env | Bin Picking | Button Press | Door Open | Drawer Open | Plate Slide | Sweep Into |
|---|---|---|---|---|---|---|
| Success Rate | 55.6% | 15.56% | 53.96% | 60.12% | 34.4% | 42.25% |

Table 4: Success rate of suboptimal checkpoints used for generating the rollout datasets.

First, we train an Oracle SAC policy to obtain $Q^*$ and $\pi^*$. To ensure low TD-error on the offline rollout dataset, we add all rollouts to the replay buffer of the SAC model before we start training. We then run SAC as usually, collecting online data, but with a sufficiently large replay buffer such that no data rollout data is overridden.

After training the policy, we estimate regret labels for *entire* segments at a time by writing the negated regret in terms of the value function and reward. We find that this lowers variance. Under deterministic

dynamics, it can be shown that:

$$-\text{regret}(\sigma) = \sum_\sigma \gamma^t A^*(s_t, a_t) = \sum_\sigma \gamma^t (Q^*(s_t, a_t) - V^*(s_t))$$

$$= \sum_\sigma \gamma^t (r(s_t, a_t) + \gamma V^*(s_{t+1}) - V^*(s_t))$$

$$= \gamma^k V^*(s_k) - V(s_0) + \sum_{t=0}^{k-1} \gamma^t r(s_t, a_t)$$

We then estimate $V^*(s) = \mathbb{E}_{a \sim \pi^*(\cdot|s)}[Q^*(s, a)]$ by evaluating the SAC $Q$ function on 64 MCMC samples from the policy. Accurately estimating the optimal advantage function over the entire sub-optimal rollout dataset was difficult. We found that this procedure lead to the best results in comparison to other means of estimating the advantage, like directly evaluating $Q^*(s, a) - V^*(s)$, using $\alpha \log \pi(a|s)$ per the bijection in maximum entropy RL, or by using MCMC rollouts of the policy from states in a segment. In the figure below we show how these options, though theoretically equivalent, do not agree in practice.

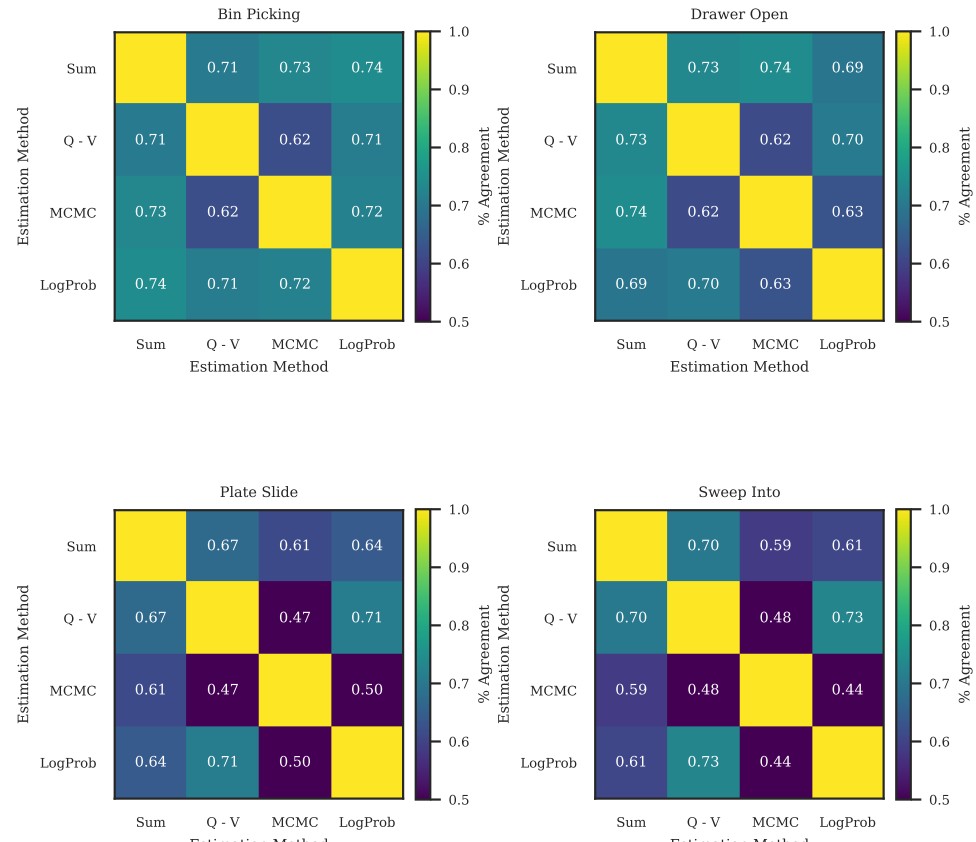

Figure 12: Comparing the agreement between different methods of estimating the advantage for generating regret based preferences. We used the sum variant because we found that it usually had the highest agreement across tasks (average of its row).

### E.3 EVALUATION

Evaluating CPL in comparison to other RL baselines can be difficult, as CPL uses only supervised learning, while P-IQL is RL based. Superivsed learning methods, like CPL can easily overfit the training data. On the other hand, off-policy RL methods like P-IQL converge to a fixed point and thus often take much longer to train before overfitting. While notable works in imitation learning (Mandlekar et al., 2021) have reported the average of the maximum performance of each seed, we find this to be a bit overly optimistic to randomness in evaluation episodes and assumes one can evaluate every checkpoint. on the other hand, offline RL works like (Kostrikov et al., 2022), report evaluation after a fixed amount of training, which can lead to overfitting for supervised approaches like CPL. We take a middle-of-the-road approach when reporting numbers in Table 1.

Every 5000 steps for state-based experiments and every 2500 steps for image-based experiments we run 25 evaluation episodes. We then average evaluation performance across eight neighboring checkpoints with a running average, totaling 200 evaluation episodes. We then average this value across seeds. Finally, we take the maximum point of the average. This exactly corresponds to the peak of the learning curves provided for all of our experiments. This evaluation procedure first averages performance over a number of checkpoints and episodes, and then averages over seeds. This maximum-of-the-average approach mitigates the over-optimism of average-of-the-maximum evaluation procedures like those used in Mandlekar et al. (2021). At the same time, it assumes that we can reasonably stop training before over-fitting begins.

For the D4RL Preference datasets results we follow the same methodology except perform 10 evaluation episodes every 5000 steps following Kim et al. (2023).

### E.4 HYPERPARAMETERS

Below we detail hyper-parameters for all methods. Note that we assumed all policies to be Gaussian. For Metaworld we used a fixed variance and thus compute the log probability $\log \pi(a|s)$ for CPL as $-||\pi(s)-a||_2^2$. For the D4RL datasets we use a diagonal gaussian distribution with a learned variance for each action dimension. For preference transformer we use the authors original codebase to re-run their results. For CPL in D4RL, we implement the CPL loss function in the codebase from An et al. (2023) which is based on Kim et al. (2023). We run results for P-IQL using the implementation from Hejna & Sadigh (2023), but remove the data augmentation to be consistent with all other methods. We left all other hyper-parameters, except for the temperatures of CPL, the same as the author's implementations.

| Hyperparameter | State | Image Sparse | Image Dense |
|---|---|---|---|
| Total Training Steps | 500k | 200k | 120k |
| Pre-training Steps (except P-IQL) | 200k | 80k | 40k |
| Batch Size | 96 | 48 | 32 |
| Segment Size | 64 | 64 | 64 |
| Actor Dropout | 0.25 | 0.5 | 0.5 |
| Architecture | [512, 512] MLP | DrQv2 | DrQv2 |

Table 5: **Common MetaWorld Hyper-parameters**. Batch size refers to the number of comparisons sampled from the dataset. So, a single batch contains $2\times$ batch size $\times$ segment size total states. We use the same batch size for all methods, even if they do not need to operate on full segments. Running so many image-based experiments is computationally expensive. Thus, for the image based experiments we lowered the batch size. Because our dense datasets had only 2.5K segments versus the 20K in our sparse datasets, we trained for fewer steps. We also lowered the batch size because the comparisons were more dense. Table 2 reports training speed for image results using the sparse datasets. We use the same architecture for all methods. Pre-training was used for CPL and variants and SFT. Pre-training steps counted towards the total training step budget, as shown in the learning curves. We found that dropout helped all methods.

| Hyperparameter | CPL | CPL (BC) | CPL (KL) | CPL for D4RL |
|---|---|---|---|---|
| Learning Rate | 0.0001 | 0.0001 | 0.0001 | 0.0001 |
| Temp $\alpha$ | 0.1 | 0.1 | 0.1 | 0.2 |
| Bias $\lambda$ | 0.5 | - | 0.75 | 0.5 |
| BC Weight $\beta$ | 0.0 | 1.0 | 0.0 | 0.0 |
| LR Schedule | - | - | 10% after pretraining | - |
| $\gamma$ | 1 | 1 | 1 | 1 |

Table 6: Hyper-parameters for CPL and variants.

| Hyperparameter | P-IQL |
|---|---|
| Expectile $\tau$ | 0.7 |
| Temperature | 0.3333 |
| Bias $\lambda$ | 0.5 |
| $\gamma$ | 0.99 |
| Reward Net Steps | 50k |
| Learning Rate | 0.0003 state, 0.0001 image |

| Hyperparameter | SFT |
|---|---|
| Learning Rate | 0.0001 |

Table 7: Hyperparameters for P-IQL and SFT for MetaWorld. We closely follow details from Kostrikov et al. (2022) and Hejna & Sadigh (2023) and tune parameters, particularly the number of reward net steps, as it is crucial for performance. Parameters not listed are left to their defaults

| Hyperparameter | PPO |
|---|---|
| Batch Size | 128 |
| Collection steps | 4096 |
| Epochs per collection | 10 |
| learning rate | 0.0003 |
| KL reward weight | 2 or 5 |
| GAE $\lambda$ | 0.95 |
| Discount $\gamma$ | 0.99 |
| Policy std-dev Limits | (0.5, 1.5) |
| Activation | Tanh |

Table 8: PPO Hyperparameters. Note that we spent a lot of time tuning PPO to try and get better results. We found that limited the standard deviation of the policy to be at least 0.5 was crucial, as well as bounding the reward values of the pretrained reward model to stay between 0 and 1. We left the network architecture the same as other baselines, but used a Tanh activation since it is suggested for PPO.

