# OpenReview forum: "Contrastive Preference Learning: Learning from Human Feedback without Reinforcement Learning"
_ICLR.cc/2024/Conference — ICLR 2024 poster_

### Official Review · Reviewer_WpDS · 2023-10-29

**Soundness:** 3 good
**Presentation:** 2 fair
**Contribution:** 2 fair
**Rating:** 6
**Confidence:** 3

**Summary:**

This work introduces Contrastive Preference Learning (CPL), a novel algorithm designed for learning optimal policies from preferences, eliminating the need to learn reward functions. CPL integrates the regret-based preference framework with the principle of Maximum Entropy, establishing a one-to-one correspondence between advantage functions and policies.
The experimental results highlight CPL's superior performance compared to SFT and offline Reinforcement Learning (P-IQL).

**Strengths:**

1. The motivation is evidently well-defined.
2. It adeptly combines theoretical analysis with empirical findings.
3. The proposed method is written in a clear and easily understandable manner.

**Weaknesses:**

This article exclusively compares CQL with offline RL, but to my knowledge, the majority of RLHF (Reinforcement Learning from Human Feedback) algorithms employ **online** RL algorithms [1]. There appears to be a fundamental distinction between these two training paradigms. Offline algorithms exclusively train the model on static datasets, whereas online algorithms train the model on the trajectories gathered by the training policies.

I strongly encourage the authors to include a baseline that trains the reward model using the dataset and subsequently employs an **online** training methodology, such as PPO. This addition is crucial to substantiate the authors' claims.


[1] Training language models to follow instructions with human feedback.

**Questions:**

What if you were to employ an online RL algorithm for the reinforcement learning experiment instead of an offline one?

---

> ### Author Response · Authors · 2023-11-17
> **Response to Reviewer WpDS -- We have added a PPO baseline.**
>
> We would like to thank the reviewer for taking the time to read our work, and were pleased to hear that it “adeptly combines theoretical analysis with empirical findings” and is “clear and easily understandable”.
>
> The reviewer states a single weakness of our work – that we do not compare to online algorithms like PPO with a learned reward model. We have run this baseline, but would like to additionally discuss our choice for not including it originally below.
>
> 1. **In sequential settings, online methods like PPO require online access to the MDP, which can be impossible**. Consider a multi-step version of the dialogue domain brought up and cited by the reviewer. At a minimum, sequential dialog data would be a sequence of at least two steps $s_1, a_1, s_2, a_2$ where $s$’s are the prompts and $a$’s are the LLMs completions. While the first prompt $s_1$ could be sampled from a fixed dataset, and the first action $a_1$ could be sampled from the LLM, the second prompt $s_2$ would need to be provided by a user as it depends on $a_1$! As PPO relies on the on-policy policy gradient, we would need humans to interact with the language model `batch_size` number of times for *every* gradient step in the sequential setting. Thus, using online methods like PPO in *sequential* domains can be infeasible. We would like to point out that the paper and RLHF domain the reviewer mentions is a contextual bandits setting (one step) and not a true sequential problem. In Appendix A we show that in this bandits setting CPL reduces to DPO, which can also outperform PPO on LLM training [1].
>
> 2. **Online methods like PPO assume the cost of data collection is essentially zero.** PPO requires rolling out the policy to estimate the gradient. In many applications, like robotics, this is very expensive and time consuming, making learning from offline data a way more attractive alternative. In fact, in that community, the offline learning approach has largely been adopted.
>
> 3. Many contemporary works on RLHF for robotics or control use only offline data [2,3,4].
>
> Finally, we ran a PPO baseline with a KL-penalty on the reward as commonly used in RLHF [5]. Before discussing results we would like to note that this comparison is unfair to CPL as:
>
> 1. The PPO baseline has access to online samples from the MDP.
>
>
> 2. The PPO baseline uses 25x the number of state-action pairs as CPL for 2.5K Dense and 4x the amount for 20K sparse. This is because PPO uses 3.84 million additional online transitions plus the original 160,000 for CPL Dense and 1.28 million for CPL Sparse).
>
>
> 3. PPO required far more hyper-parameter tuning. We had to bound the learned rewards, bound the policy standard deviation, and aggressively tune the KL-divergence.
>
> Despite all of this, we found PPO to be extremely unstable and often underperformed CPL in 4 of the environments. Note that we verified the performance of our PPO implementation on standard Benchmarks (Hopper-v2, exceeds performance [here](https://spinningup.openai.com/en/latest/spinningup/bench.html)). Our full results are in Table 1 for two different values of the reward KL-penalty, 2 and 5, and learning curves for PPO are in Appendix C. We have copied them here for convenience:
>
> *2.5K Dense Results*
> | | Bin Picking | Button Press | Door Open | Drawer Open | Plate Slide | Sweep Into |
> |---------|-----------------|-----------------|-----------------|-----------------|-----------------|------------------|
> | PPO KL2 | 83.7 $\pm$ 3.7 | 22.7 $\pm$ 1.9 | 79.3 $\pm$ 1.2 | 66.7 $\pm$ 8.2 | 51.5 $\pm$ 3.9 | 55.3 $\pm$ 6.0 |
> | PPO KL5 | 83.2 $\pm$ 2.4 | 18.7 $\pm$ 3.1 | 68.8 $\pm$ 3.0 | 59.0 $\pm$ 2.5 | 43.0 $\pm$ 2.2 | 50.7 $\pm$ 15.4 |
> | CPL | 80.0 $\pm$ 2.5 | 24.5 $\pm$ 2.1 | 80.0 $\pm$ 6.8 | 83.6 $\pm$ 1.6 | 61.1 $\pm$ 3.0 | 70.4 $\pm$ 3.0 |
>
> *20K Sparse Results*
>
> |         | Bin Picking     | Button Press    | Door Open       | Drawer Open     | Plate Slide     | Sweep Into      |
> |---------|-----------------|-----------------|-----------------|-----------------|-----------------|-----------------|
> | PPO KL2 | 68.0 $\pm$ 4.3 | 24.5 $\pm$ 0.8 | 82.8 $\pm$ 1.6 | 63.2 $\pm$ 6.6 | 60.7 $\pm$ 4.2 | 58.2 $\pm$ 0.6 |
> | PPO KL5 | 71.1 $\pm$ 6.7 | 23.0 $\pm$ 1.8 | 71.3 $\pm$ 2.1 | 64.7 $\pm$ 7.7 | 48.0 $\pm$ 3.9 | 53.7 $\pm$ 1.7 |
> | CPL     | 83.2 $\pm$ 3.5 | 29.8 $\pm$ 1.8 | 77.9 $\pm$ 9.3 | 79.1 $\pm$ 5.0 | 56.4 $\pm$ 3.9 | 81.2 $\pm$ 1.6 |
>
> PPO faces a number of challenges in the sequential domain: the variance of policy gradients and value estimation increases with horizon, and instability of the KL-divergence penalty for continuous policies. Again, in 4/6 environments CPL attains better performance than PPO using 1/4 of the data.
>
> As we have addressed the reviewers' only stated concern with the draft through this experiment, we would kindly request that the reviewer reconsider their score.
>
> [1] Rafailov, Sharma, Mitchell, Direct Preference Optimization. NeurIPS 2023
>
> [2] Kim et al. Preference Transformer. ICLR 2023.

---

> > ### Author Response · Authors · 2023-11-17
> > **Response to reviewer WpDS (references continued)**
> >
> > [3] Hejna et al. Inverse Preference Learning. NeurIPS 2023
> >
> > [4]  Kang et al. Offline Preference-guided Policy Optimization, ICML 2023
> >
> > [5] Ziegler, Stiennon et al. Fine-Tuning Language Models from Human Preferences. 2020

---

> > > ### Author Response · Authors · 2023-11-21
> > >
> > > As the discussion period is ending soon, it would be great to hear from the reviewer as we have addressed their stated weakness of the draft by running a PPO baseline.
> > >
> > > Thank you!
> > > Authors

---

> ### Comment · Reviewer_WpDS · 2023-11-22
>
> Thanks for your response!
>
> I am very grateful to the authors for conducting experiments on online reinforcement learning.
>
> I requested such experiments because the authors extensively emphasize the advantages of CPL for RLHF. Without clarifying that the entire setting of the article is offline, I believe an online baseline is necessary.
>
> For the experiments of PPO, it seems PPO-KL2 outperforms PPO-KL5 in almost all experiments. I understand that the authors may have time constraints to tune the hyper-parameters, and I believe that online algorithms also have great potential.
>
> Overall, the author did address my concern, and I would like to increase my score.

---

> > ### Author Response · Authors · 2023-11-22
> > **Thank you!**
> >
> > We would like to thank the reviewer for engaging in discussion and are pleased to hear we have addressed their concern.
> >
> > Briefly on hyper parameters -- we did spend lots of time tuning, which was difficult when trying to optimize across all 6 environments. We included KL Coefficient 5 for completeness, but will focus on coefficient 2 in the text as it performs best in our experiments.
> >
> > Note that we also tried lower KL penalty coefficients (like 1), but they performed far worse in our parameter sweep:
> >
> > *20K Sparse Results*
> >
> > |         | Bin Picking      | Button Press    | Door Open       |
> > |---------|------------------|-----------------|-----------------|
> > | PPO KL1 | 58.7 $\pm$ 10.0 | 19.2 $\pm$ 3.1 | 55.5 $\pm$ 3.5 |
> > | PPO KL2 | 68.0 $\pm$ 4.3  | 24.5 $\pm$ 0.8 | 82.8 $\pm$ 1.6 |
> > | PPO KL5 | 71.1 $\pm$ 6.7  | 23.0 $\pm$ 1.8 | 71.3 $\pm$ 2.1 |

---

### Official Review · Reviewer_moLC · 2023-11-01

**Soundness:** 4 excellent
**Presentation:** 3 good
**Contribution:** 4 excellent
**Rating:** 8
**Confidence:** 4

**Summary:**

The authors address critical aspects of the PBRL framework, with a specific emphasis on the optimization challenges in the RL phase. To solve this problem, the authors introduce a novel approach called Contrative Preference Learning (CPL). This method leverages a regret-based model of human preferences, from which a contrastive objective is derived with the principle of maximum entropy. This approach bypasses the need for reward learning and RL, instead directly learn the policy through a supervised learning paradigm. To evaluate the effectiveness of CPL, the authors conducted experiments within the offline PbRL setting, comparing it against strong baselines in terms of the success rate across distinct tasks in the Metaworld domain. The experimental results show that CPL outperforms baselines with less runtime and smaller model size. The primary contribution of this work lies in the conversion of the traditional two-phase PbRL framework into a novel paradigm capable of directly learning the policy with a new contrastive objective.

**Strengths:**

This work focuses on addressing critical challenges in PbRL. It is well-motivated and accompanied by a clear and thorough discussion of existing issues within both the reward learning and RL phases.

The proposed CPL bypasses the need for reward learning and RL by optimizing a supervised objective, enabling it to learn policy from offline high-dimensional suboptimal data. Moreover, it can be applied to arbitrary MDPs. I feel this approach can be seen as a counterpart to DPO, as discussed by the authors in the paper—one for NLP tasks with LLMs and the other for continuous control tasks. This work has the potential to make a significant impact in the community, and I am eager to see how CPL performs in broader applications.

Generally, the organization and presentation of the content are well-structured, facilitating ease of reading and comprehension. The authors provide comprehensive theoretical proofs that make the work sound. The experimental results are impressive in terms of runtime, model size, and performance. In the limitation section, I appreciate the authors acknowledge the imperfections of the human model and raise considerations regarding the application of this approach to online human feedback.

**Weaknesses:**

Please see Questions.

**Questions:**

1. I still have questions regarding regret-based preference model. I agree with the authors that the regret-based preference model makes more sense when we consider the hand-engineered example in section 2. However, when we talk about data collection with a real human, the human labeler would have a preference over two trajectory segments. This implies the existence of underlying dense rewards that explain the human's preferences. In such cases, I feel that the key issue lies in the hand-engineered reward is incorrect (i.e., reward design issue) in your example, rather than in the issues of the reward-based preference model.

Therefore, when we consider experiments with real humans and apply the reward-based preference model, could it also perform effectively? Is it possible that the learned reward captures the regret information to a large extent? Please correct me if I have misunderstood.

2. Despite considering the model complexity of CPL, the results are promising. In terms of feedback efficiency, does CPL require more human preference data compared to the conventional two-phase PbRL framework in order to perform well? This is especially relevant considering the Metaworld tasks in the experiments, where obtaining dense data could be challenging if collected from real humans.

3. In the experiments, the authors pretrain the model with behavior cloning. To what extent does this pretraining phase impact the model's final performance? Does P-IQL also have this pretraining phase?

4. Similar to DPO, CPL employs a supervised learning framework without reward learning and RL. Does it potentially lose the generalization power of RL?

---

> ### Author Response · Authors · 2023-11-17
> **Response to Reviewer moLC (1/2)**
>
> We were excited to hear that the reviewer found our work to be “well-motivated”, “clear”, and that it “has the potential to make a significant impact in the community”. The reviewer's comments were largely based on a set of additional questions we seek to answer here.
>
> **The Regret Preference Model**
>
> The regret preference model states that human *judgements* are better captured by an optimal advantage function, and does not make any claims about what underlying reward functions humans use. Going back to the example in Section 2, we did not mean to claim that the human would intend to use a sparse reward function – a user providing preferences preferring movements towards a goal could have a number of different underlying reward functions, many of which could yield the same optimal policy and optimal advantage function.
>
> Though the focus of our paper is not the validity of the regret preference model, for which we would refer the reader to [1], we do provide a number of additional insights on the regret preference model.
>
> 1. *Theoretical*: In Appendix A, we prove that the optimal advantage is just a shaped version of the reward function that results in the same optimal policy. The reviewer suggests that humans use dense rewards to make decisions, and Corollary 1 shows exactly that the advantage is a dense reward function.
>
> 2. *Empirical*: In the new Section 4.4, we show that CPL works on real human data, indicating that the regret-model holds on real human data.
>
> Again we would encourage the reader to examine [1], as it contains a number of additional examples (like stochastic environments) and experimental results which support the validity of the regret-preference model.
>
> **Does CPL require more human preference data in comparison to two-phase PBRL**
>
> To demonstrate that CPL is effective with limited real-human preference data we perform additional evaluations on the PBRL benchmarks from [2]. These benchmarks use only 100 or 500 real human preferences for expert or replay datasets from D4RL respectively.
>
> We showed in Section 4.3 that CPL performs better with dense preference data. To overcome this, we use a logistic regression model to predict which of two segments a user prefers from the preference data, and use it to relabel a larger offline RL dataset. Critically, this logistic regression model is not a reward or advantage model, as it predicts a single value for the *entire* segment.
>
> We find that CPL performs similarly to baselines on 2/4 datasets, outperforms in 1, and performs worse in 1. Yet, CPL remains far simpler.
>
>
> **Pretraining**
>
>
> We have included results on Metaworld in state without any pre-training, and find that CPL performs similarly – no pretraining performs better in some cases and pretraining performs better in others. This ablation has been included in Appendix C and is copied below as well.
>
> |                   | Bin Picking     | Button Press    | Door Open       | Drawer Open     | Plate Slide     | Sweep Into      |
> |-------------------|-----------------|-----------------|-----------------|-----------------|-----------------|-----------------|
> | CPL               | 80.0 $\pm$ 2.5 | 24.5 $\pm$ 2.1 | 80.0 $\pm$ 6.8 | 83.6 $\pm$ 1.6 | 61.1 $\pm$ 3.0 | 70.4 $\pm$ 3.0 |
> | CPL (No Pretrain) | 81.0 $\pm$ 1.5 | 30.0 $\pm$ 2.7 | 76.8 $\pm$ 2.1 | 88.7 $\pm$ 1.9 | 50.8 $\pm$ 3.5 | 80.0 $\pm$ 2.5 |
>
>
> This ablation shows that the use of the conservative regularizer in CPL can help it learn effectively even without pre-training. Note that our results on D4RL (Table 3) also do not use any pre-training.
>
>
> We did not use pre training for P-IQL for two reasons:
>
>
> 1. P-IQL uses a weighted BC objective ($\min \mathbb{E} [ e^{Q-V} \log \pi]$ and thus already has a loss function very similar to BC.
> 2. In IQL value learning is completely detached from policy learning, and thus pretraining $\pi$ does not improve estimates of $Q$ and $V$.
>
> **Does CPL lose the generalization power of RL**
>
> We are not 100% certain what the reviewer is referring to by the “generalization power of RL”.
>
> To our best understanding, there are two types of generalization potentially at play in DeepRL: 1) the generalization of neural networks and 2) RLs ability to use the ground-truth reward function to extrapolate in new parts of the state space. For 1), CPL still has the generalization power of neural networks. For 2), we would like to point out that this does not necessarily apply in RLHF, as the accuracy of the reward model is dependent on the preference data. Thus, RLHF methods that use an explicit reward model at the end of the day, still rely on the generalization of supervised reward learning. CPL is just cutting out the middleman.
>
> This approach has already shown to be effective for LLMs [3]. By removing the need for extra components, we believe CPL will help scale to larger datasets/models where better generalization is typically found.
>
> continued...
>
> [1] Knox et al.  https://arxiv.org/abs/2206.02231

---

> > ### Author Response · Authors · 2023-11-17
> > **Response to Reviewer moLC (2/2)**
> >
> > If the reviewer was referring to the ability of a reward model to be applied to unlabeled data, we show in our D4RL experiments that CPL can also take advantage of unlabeled data by using a logistic-regression based relabeling model.
> >
> > [1] Knox et al. Models of Human Preference for Learning Reward Functions. 2023 https://arxiv.org/abs/2206.02231
> >
> > [2] Kim et al. Preference Transformer. ICLR 2023.
> >
> > [3] Rafailov, Sharma, Mitchell, Direct Preference Optimization. NeurIPS 2023

---

> > > ### Comment · Reviewer_moLC · 2023-11-18
> > > **Post Rebuttal Comments**
> > >
> > > Thanks for the authors' responses, and I appreciate the inclusion of the additional experiments, which have addressed my concerns and clarified a few questions. After carefully reviewing both the authors' responses and other reviewers' comments, I am inclined to maintain my score and recommend accepting the paper.

---

### Official Review · Reviewer_Vhgt · 2023-11-03

**Soundness:** 4 excellent
**Presentation:** 4 excellent
**Contribution:** 3 good
**Rating:** 8
**Confidence:** 2

**Summary:**

This paper presents a Contrastive Preference Learning (CPL) framework for learning optimal policies from human preference data without learning the reward function. Specifically, the paper models human preferences using the advantage function and proposes a general loss function for learning policies. The loss function ensembles the contrastive learning objective and can be optimized directly without learning a reward function. As a result, the method can scale to high-dimensional environments and sequential RLHF problems (i.e., beyond contextual bandits). Theoretically, by optimizing the loss function, CPL provably converges to the optimal policy of the underlying max-entropy RL problem. The paper tests one instantiation of the CPL framework and shows its promising performance in practice.

**Strengths:**

- The proposed algorithmic framework is novel and elegant. The motivation for the problem is clear.

- The method is scalable without the use of RL.

- The experimental results are adequate.

- The paper is very well-written and easy to follow.

**Weaknesses:**

I did not identify any noticeable weaknesses.

**Questions:**

Since the CPL loss function has a super elegant form, is it possible to derive finite sample analysis for learning a near-optimal policy like [1]?

[1] Zhu et al., Principled Reinforcement Learning with Human Feedback from Pairwise or K-wise Comparisons, ICML 2023

---

> ### Author Response · Authors · 2023-11-17
> **Thank you for the review -- let us know if you have questions!**
>
> We would like to thank the reviewer for their time in reviewing our work. We are happy to hear that our algorithm was “novel and elegant”, and that it’s “motivation… is clear”.
>
> As the reviewer has pointed out no weaknesses, we will just mention the following revisions to the draft:
>
> 1. Additional PPO baseline. We have modified Table 1 with an additional PPO baseline. Note that as PPO is an online RL method, it requires 4x the amount of state-action data as CPL plus online interactions with the MDP. Despite that, it still underperforms in many settings and exhibits low variance.
>
> 2. Additional Benchmark with Real Human Feedback. We have added additional experiments on the D4RL benchmark using only 100 to 500 real-human queries.
>
> 3. Additional Theoretical revisions. We have heavily modified the appendix and Section 3.3 including a new proposition on CPLs regularization with finite data.
>
> 4. Additional ablations on pretraining in Appendix C.

---

### Official Review · Reviewer_NpZJ · 2023-11-06

**Soundness:** 3 good
**Presentation:** 3 good
**Contribution:** 3 good
**Rating:** 6
**Confidence:** 4

**Summary:**

This paper presents Contrastive Preference Learning (CPL), an algorithm to learn optimal policies from preferences without learning exlicitly a reward function which is commonly done in RLHF scenario. This circumvents the issue of having an underoptimized/overoptimized reward model. The authors then show the performance of CPL for MetaWorld benchmark.

**Strengths:**

1. learning a reward model from human preferences has flaws. And on top of that, using RL to optimize for this reward model can sometimes lead to poor performance. This paper solves this issue by not having a reward model.
2. CPL has supervised objectives so it is scalable
3. The proposed algorithm is generic

**Weaknesses:**

1. In my experience, learning a "good" reward model and then doing RL always outperforms offline RL algorithms. The authors only compare it with IQL and not with methods with reward models to highlight more
2. The authors claim that the method is generic but then it is only applied to MetaWorld benchmark. The RLHF scenario is much more interesting in aligning language models with human feedback.

**Questions:**

How does CPL compare with RLHF for language models scenario?
How does CPL compare with other baselines, which may or may not have reward models

---

> ### Author Response · Authors · 2023-11-17
> **Response to reviewer NpZJ**
>
> Thank you for taking the time to review our work. We were happy to hear that the reviewer found our work to be “generic”, “scalable”, and overcoming challenges in reward optimization. The reviewer had two primary questions about our work, first concerning our baselines and second concerning our benchmarks.
>
> **Baselines**
>
> The reviewer asked us to compare CPL to methods that learn a reward model.
>
> 1. We would graciously like to point out that P-IQL does indeed learn a reward model first, which could then be applied with any RL Algorithm. We chose to use IQL since it is generally a very  high-performing offline RL method [1] and has been used as a standard in many prior RLHF for control works [2, 3].
>
> 2. We have additionally compared CPL to PPO with a learned reward function based on comments from Reviewer WpDS.  We would like to point out that this comparison is *unfair* to CPL, as PPO requires additional online data to estimate the policy gradient. While in bandits settings (like LLMs) this can be done without access to the MDP, in sequential settings (like control, or 2+ steps of dialogue) we need to interact with the MDP to get full trajectories, which can be expensive (for ex running a robot in the real world). Despite the fact that PPO uses online data and 4x the number of state-action pairs, it is highly unstable in comparison to CPL and often underperforms it by a large margin. We have included full results in Table 1.
>
> 3. Our results are consistent with other recent works [3, 4] which have also shown that learning an explicit reward model is unnecessary to achieve good performance.
>
> **Benchmarks**
>
> 1. The reviewer mentioned that RLHF domains with LLMs would be interesting to explore. We whole-heartedly agree that RLHF with LLMs is a great possible application of CPL. Our work, however is about settings with *sequential* data  in particular, which in dialogue would constitute multiple turns of interaction with a human user. To our knowledge there were no such publicly available RLHF datasets with 2+ turns of dialogue at the time of submission. We believe such an investigation is a large undertaking, enough to warrant its own publication.
>
> 2. We would kindly like to push back against the notion that robotics domains are less interesting. We chose to evaluate in robotics domains precisely because they capture the sequential nature of CPL, and show that our method applies to general MDPs, instead of the  simpler contextual bandit setting used in standard RLHF with LLMs.
>
> 3. We have included additional benchmark results on more robotics domains using only 100 or 500 queries of real-human feedback [2]. We hope this serves as another datapoint for the applicability of both CPL and the regret-preference model. Please see Table 3 in the paper for results, or the general response for results. We find that CPL is able to perform similarly to other methods while being vastly simpler as it requires no Q-learning. CPL also exhibits lower variance.
>
> We hope the reviewer can consider these additional empirical results, as well as our revised theoretical contributions when assessing our work.
>
> [1] Kosterikov et al. Implicit Q-Learning ICLR 2022.
>
> [2] Kim et al. Preference Transformer. ICLR 2023.
>
> [3] Hejna et al. Inverse Preference Learning. NeurIPS 2023
>
> [4] Rafailov, Sharma, Mitchell, Direct Preference Optimization. NeurIPS 2023

---

### Author Response · Authors · 2023-11-17
**Thank you for the reviews & additional baselines, benchmarks, and theory**

We would like to thank all the reviewers for taking the time to review our work. Though we respond to each reviewer individually below, we outline all general improvements to the draft here.

### **1. Comparisons to PPO**
We have added additional comparisons to PPO with a reward KL penalty [1] in Table 1 from state. Note that this is not a fair comparison as PPO uses 25x the number of state-action pairs as CPL 2.5K Dense and 4x that of CPL 20K Sparse because it requires online rollouts. Despite this, PPO remains unstable and largely underperforms CPL. In the tables below, the number next to PPO is the coefficient of the KL-reward penalty with the pretrained policy.

**2.5K Dense Results**
| | Bin Picking | Button Press | Door Open | Drawer Open | Plate Slide | Sweep Into |
|---------|-----------------|-----------------|-----------------|-----------------|-----------------|------------------|
| PPO KL2 | 83.7 $\pm$ 3.7 | 22.7 $\pm$ 1.9 | 79.3 $\pm$ 1.2 | 66.7 $\pm$ 8.2 | 51.5 $\pm$ 3.9 | 55.3 $\pm$ 6.0 |
| PPO KL5 | 83.2 $\pm$ 2.4 | 18.7 $\pm$ 3.1 | 68.8 $\pm$ 3.0 | 59.0 $\pm$ 2.5 | 43.0 $\pm$ 2.2 | 50.7 $\pm$ 15.4 |
| CPL | 80.0 $\pm$ 2.5 | 24.5 $\pm$ 2.1 | 80.0 $\pm$ 6.8 | 83.6 $\pm$ 1.6 | 61.1 $\pm$ 3.0 | 70.4 $\pm$ 3.0 |

**20K Sparse Results**

|         | Bin Picking     | Button Press    | Door Open       | Drawer Open     | Plate Slide     | Sweep Into      |
|---------|-----------------|-----------------|-----------------|-----------------|-----------------|-----------------|
| PPO KL2 | 68.0 $\pm$ 4.3 | 24.5 $\pm$ 0.8 | 82.8 $\pm$ 1.6 | 63.2 $\pm$ 6.6 | 60.7 $\pm$ 4.2 | 58.2 $\pm$ 0.6 |
| PPO KL5 | 71.1 $\pm$ 6.7 | 23.0 $\pm$ 1.8 | 71.3 $\pm$ 2.1 | 64.7 $\pm$ 7.7 | 48.0 $\pm$ 3.9 | 53.7 $\pm$ 1.7 |
| CPL     | 83.2 $\pm$ 3.5 | 29.8 $\pm$ 1.8 | 77.9 $\pm$ 9.3 | 79.1 $\pm$ 5.0 | 56.4 $\pm$ 3.9 | 81.2 $\pm$ 1.6 |




### **2. D4RL Benchmark with Real-Human Feedback**

We compare CPL to baselines on the RLHF D4RL benchmark from [2] which uses only 100 or 500 real human user preferences. To adapt CPL to this setting with additional offline data, we train a logistic regression model to relabel the offline dataset (note that this is not a reward function, as it makes a single prediction from a whole segment). We find that CPL does best on Hopper Expert, performs worse on Hopper Replay, and performs comparably on the remaining Walker datasets, despite being simpler and requiring fewer hyperparameters. CPL also exhibits lower variance in 3 of 4 datasets.

|                 | Walker-Med-Exp   | Walker-Med-Replay | Hopper-Med-Exp   | Hopper-Med-Replay |
|-----------------|------------------|-------------------|------------------|-------------------|
| P-IQL           | 99.9 $\pm$ 6.2  | 71.6 $\pm$ 5.9    | 88.6 $\pm$ 3.6  | 60.2 $\pm$ 20.6  |
| PrefTransformer | 110.2 $\pm$ 0.8 | 76.6 $\pm$ 3.2   | 86.7 $\pm$ 6.8  | 81.0 $\pm$ 10.3  |
| CPL             | 109.2 $\pm$ 0.2 | 48.3 $\pm$ 3.7   | 109.1 $\pm$ 0.7 | 72.4 $\pm$ 3.1   |



### **3. Theory**
We generally overhauled Section 3.2 on regularization and Appendix A with additional theoretical results on the underspecification of preference models with finite data and CPL’s conservative regularizer.


### **4. Pre-training Ablation**

We run ablations on CPL with and without pre-training (Appendix C), and find that CPL often-times can attain higher performance without pre-training due to the use of our conservative regularizer.

Core changes to the paper are highlighted in blue in the PDF. Thank you for your consideration.



**References**

[1] Ziegler, Stiennon et al. Fine-Tuning Language Models from Human Preferences. 2020

[2] Kim et al. Preference Transformer. ICLR 2023.

[3] Hejna et al. Inverse Preference Learning. NeurIPS 2023

---

### Public Comment · ~Yachen_Kang1 · 2023-11-19

Dear Authors,

I have read your paper titled "Contrastive Preference Learning: Learning from Human Feedback without Reinforcement Learning" with great interest. I would like to provide a public comment and suggest the inclusion of Kang et al.'s paper titled "Offline Preference-guided Policy Optimization" at ICML 2023 as a baseline for comparison.

Both papers seem to utilize contrastive learning for preference modeling and achieve policy optimization without independently learning a reward function. Additionally, both papers make use of offline datasets for training. Given these similarities,  I believe it would be beneficial for the authors to include OPPO as a baseline or, at the very least, address and provide a more comprehensive explanation in the main text as to why they did not consider or compare their approach with Kang et al.'s work. In my observation, the authors' description of the omission of this baseline in the paper seems inconsistent with the original work.

---

> ### Author Response · Authors · 2023-11-19
>
> We would like to thank the authors of OPPO for finding our work interesting, and note that we have already included a citation to their paper.
>
> First, we are open to expanding the discussion of OPPO in our work. OPPO's objective works in a rather different paradigm than CPL's as it uses HIM rather than a theoretical derivation from RL principles. Currently, the popular paradigm for RLHF -- reward learning followed by (offline) RL -- more closely follows the latter and thus we focused our experimental efforts on comparisons with these methods which CPL seeks to replace. OPPO is a bit more complicated -- it first learns a preference model, estimates an optimal $z^*$, then learns a conditioned policy.
>
> Second, we are open to including OPPO as baseline in Table 3, however we would request that the authors of OPPO give us sufficient time to ensure such a comparison is fair. OPPO uses a BERT-like preference model, a non-markovian transformer policy, and varies hyper-parameters across environments (Table 7 & Table 8 in Appendix). CPL and our baselines on the other hand use only simple MLP/CNN markovian policies and the same hyper parameters across domains. We are going to start work on an OPPO baseline using your offical codebase, but would like to first normalize all decision decisions and the evaluation procedure to ensure a fair comparison.
>
> Thanks,
> CPL Authors

---

> ### Public Comment · ~Yachen_Kang1 · 2023-11-20
>
> Dear Authors,
>
> I appreciate the author's attention to the public comment and their prompt response. The reasons mentioned by the author seem reasonable to me.
>
> It is encouraging to see further work being done in this direction, with the proposal and discussion of new methods. However, due to the constant emergence of new methods, I believe readers would prefer a comprehensive and objective understanding of the development path of relevant methods, as well as the specific similarities and differences of new methods. This would allow for a more essential understanding of the contribution of the paper.

---

### Meta-Review · Area_Chair_aBN9 · 2023-12-11

**Metareview:**

This paper introduces Contrastive Preference Learning (CPL), an innovative algorithm developed to derive optimal policies from preferences without the explicit learning of a reward function typically found in RLHF scenarios. Specifically, it capitalizes on the advantage function to represent human preferences and offers a universal loss function for policy learning. This method demonstrates scalability in high-dimensional environments and effectively addresses sequential RLHF issues (beyond the scope of contextual bandits). Theoretically, upon optimization of the loss function, the CPL algorithm is proven to converge towards the optimal policy of the inherent max-entropy RL problem. The formidable performance of one implementation of the CPL framework in a practical setting further corroborates its efficacy. The overall rating of this work is actually positive. AC appreciates the authors effort in the rebuttal stage.

**Justification For Why Not Higher Score:**

N/A

**Justification For Why Not Lower Score:**

N/A

---

### Decision · Program_Chairs · 2024-01-16

Accept (poster)